# Integrated assessment modeling of a zero-emissions global transportation sector

Simone Speizer [1], Jay Fuhrman [1], Laura Aldrete Lopez [2], Mel George [3], Page Kyle [1], Seth Monteith[2] & Haewon McJeon [4] ✉

Currently responsible for over one fifth of carbon emissions worldwide, the transportation sector will need to undergo a substantial technological transition to ensure compatibility with global climate goals. Few studies have modeled strategies to achieve zero emissions across all transportation modes, including aviation and shipping, alongside an integrated analysis of feedbacks on other sectors and environmental systems. Here, we use a global integrated assessment model to evaluate deep decarbonization scenarios for the transportation sector consistent with maintaining end-of-century warming below 1.5 °C, considering varied timelines for fossil fuel phase-out and implementation of advanced alternative technologies. We highlight the leading low carbon technologies for each transportation mode, finding that electrification contributes most to decarbonization across the sector. Biofuels and hydrogen are particularly important for aviation and shipping. Our most ambitious scenario eliminates transportation emissions by mid-century, contributing substantially to achieving climate targets but requiring rapid technological shifts with integrated impacts on fuel demands and availability and upstream energy transitions.

Due to its dependence on oil as a fuel and the distributed nature of resulting emissions, the transportation sector is one of the most challenging sectors to fully decarbonize[1–3]. The sector has been found to respond more slowly than other sectors to the imposition of climate policy such as a carbon tax, leading to relatively large residual emissions that must be offset with $CO_2$ removal under net zero emissions goals[4–8]. Substantial progress has been made in passenger road transport with the increasing development and adoption of electric light-duty vehicles[9–11], and batteries and hydrogen fuel cells offer potential for road freight vehicles[2,12]. However, low emissions technologies for maritime shipping and aviation are nascent owing to the large quantities of energy-dense fuel required for their long-distance travel[2,10,12,13]. Efforts to rapidly decarbonize these modes are further complicated by long vehicle lifespans and, for aviation in particular, the need for a dependable fuel supply that can ensure safety and sustain operations at cruising altitude[2,10]. Aviation and shipping together produced about a quarter of global transportation emissions in 2015, with this share projected to increase, especially under stringent climate mitigation scenarios that induce rapid integration of zero emissions road vehicles[13–15]. Thus, achieving deep decarbonization in the transportation sector will depend on the development of emerging low carbon technologies for aviation and shipping.

Despite their importance for meeting climate goals, few studies have analyzed mitigation strategies for the full transportation sector, including aviation and shipping, in the context of global, economy-wide decarbonization. While there are a variety of industry scenarios and analyses for aviation and shipping decarbonization in isolation[13,16–22], these do not employ an integrated framework that considers the connections between the transportation sector and other sectors and environmental systems. Such a framework is important as it allows for an assessment of the upstream emissions from transportation as well as the implications of transportation fuel

[1]Joint Global Change Research Institute, Pacific Northwest National Laboratory, College Park, MD, USA. [2]ClimateWorks Foundation, San Francisco, CA, USA. [3]Center for Global Sustainability, University of Maryland, College Park, MD, USA. [4]Graduate School of Green Growth & Sustainability, Korea Advanced Institute of Science and Technology, Daejeon, Republic of Korea. ✉e-mail: hmcjeon@kaist.ac.kr

use for land use and fuel availability for other sectors. Studies of transportation decarbonization that do incorporate interactions with the full economy are mostly restricted to a regional level[4,11,14,23–28], aggregate the entire transport sector[29], focus primarily on road vehicles[1,5,11,26,28,30], or only observe limited mitigation in aviation and shipping[8,11,15,31–35], often due to a lack of modeled options for technologies that could decarbonize these modes. Other work has assessed deep decarbonization in aviation and freight transport with economy-wide modeling, but without an evaluation of the relative contributions of different low carbon fuels to this decarbonization[2,6,36]. Recent commitments to ambitious net zero carbon emissions targets by aviation and shipping trade associations and national and international organizations[16,37–40] further highlight the need for a better understanding of how aggressive emissions reductions by these modes can be achieved alongside decarbonization measures in other transportation modes and sectors.

To take up this question, we use the Global Change Analysis Model (GCAM) version 6.0 to examine decarbonization scenarios for the entire global transportation sector consistent with limiting end-of-century warming to 1.5 °C. GCAM is a global integrated assessment model that links the energy system, water, land, and the climate, and has a highly detailed treatment of the transportation sector[6,8,30,32,33,36]. We consider decarbonization strategies for all modes of passenger and freight transportation represented in GCAM (Supplementary Fig. 1) but focus particularly on analyzing pathways for aviation and shipping. We examine high, medium, and low ambition scenarios, with higher degrees of ambition corresponding to higher levels of elimination of traditional fossil fuels and adoption of advanced alternative technologies (Table 1). The high ambition scenario considers a complete phase-out of fossil fuels from the transportation sector by 2050. The medium scenario delays the elimination of fossil fuels in transportation until 2100, while the low scenario does not impose sector-specific requirements for the decarbonization of transport. These varied degrees of ambition allow for an evaluation of the integrated system responses resulting from different levels of realization of climate goals in the transportation sector, spanning the range of the most aggressive decarbonization targets that have been set by industry leaders[37–40] (high scenario) to the minimum goals agreed upon by international organizations[41,42] (medium and low scenarios).

Along with these technological shifts, we also assume reduced demand for transportation services, increased ridesharing, and increased public transportation use. We hold these demand-side assumptions constant across our three decarbonization scenarios to isolate the effects of the varied level of technological ambition. We pair the changes in the transportation sector with additional technological and behavioral changes across other sectors of the economy that facilitate greenhouse gas mitigation, in line with the 1.5 °C scenario from Gambhir et al.[43] and the 1.5 °C "sectoral strengthening" scenario from Fuhrman et al.[44] (see Methods for details)[43,44]. These shifts are aligned with or surpass the "sustainability" scenario (SSP1) from the Shared Socioeconomic Pathways (SSPs)[45,46]. We employ a global carbon emissions constraint that ensures end-of-century warming is below 1.5 °C and that is consistent with the Sixth Assessment Report (AR6) of the Intergovernmental Panel on Climate Change (IPCC) emissions scenarios Category 1 (Limit warming to 1.5 °C (>50%) with no or limited overshoot)[47]. For comparison with our decarbonization scenarios, we also consider a reference scenario corresponding to a continuation of current trends and technological developments and in which no climate policy is implemented (Table 1). We maintain the same population and GDP growth assumptions (both following SSP1) in the reference scenario as in the decarbonization scenarios (Supplementary Fig. 2) to avoid confounding effects of varied socio-economic pathways on transportation and resource demands across scenarios.

**Table 1 | Scenario matrix**

| Scenario | Shipping and aviation | Passenger cars and trucks, buses | Freight trucks | Passenger and freight rail | Full transport sector | Other sectors | Population and GDP |
|---|---|---|---|---|---|---|---|
| Reference (no climate policy) | Base GCAM | Base GCAM | Base GCAM | Base GCAM | Base GCAM | Base GCAM | SSP1 |
| 1.5 °C low transport tech | Lower levels of electric and hydrogen-based technologies. No biofuels/e-fuels requirement | Slower integration of electric and hydrogen-based technologies. Fossil fuel-based sales continue | Slower integration of electric and hydrogen-based technologies. Fossil fuel-based sales continue | Slower integration of electric and hydrogen-based technologies | Reduced demand, increased ridesharing, increased public transit preference | Behavioral and technological changes to facilitate emissions reductions | |
| 1.5 °C medium transport tech | Medium levels of electric and hydrogen-based technologies. Fossil fuels replaced by biofuels/e-fuels by 2100 | Medium integration of electric and hydrogen-based technologies. No new fossil fuel-based sales by 2050 | Medium integration of electric and hydrogen-based technologies. No new fossil fuel-based sales by 2060 | Medium integration of electric and hydrogen-based technologies. No new fossil fuel-based sales by 2090 | | | |
| 1.5 °C high transport tech | High levels of electric and hydrogen-based technologies. Fossil fuels replaced by biofuels/e-fuels by 2050 | Fast integration of electric and hydrogen-based technologies. No new fossil fuel-based sales by 2030 | Fast integration of electric and hydrogen-based technologies. No new fossil fuel-based sales by 2035 | Fast integration of electric and hydrogen-based technologies. No new fossil fuel-based sales by 2050 | | | |

Describes the levels of technological ambition for each of the transportation modes, as well as the assumptions that apply to the full transportation sector and other sectors, in each scenario. Note that the category of passenger cars and trucks includes all passenger light duty vehicles. Also note that for all modes except aviation, hydrogen-based technologies correspond to fuel cell electric vehicles; for aviation, hydrogen combustion turbines are modeled.
GCAM Global Change Analysis Model, GDP gross domestic product, SSP1 Shared Socioeconomic Pathway 1.

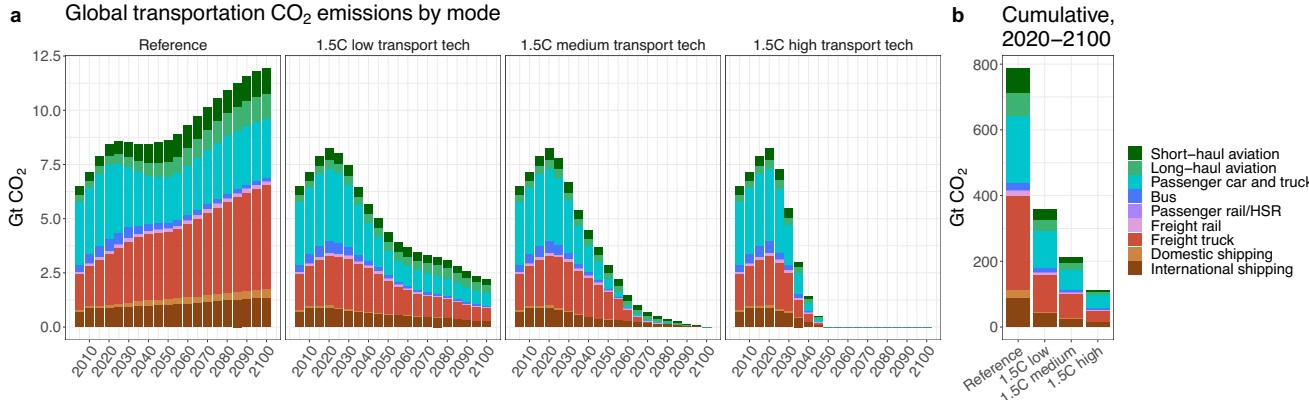

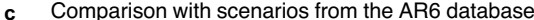

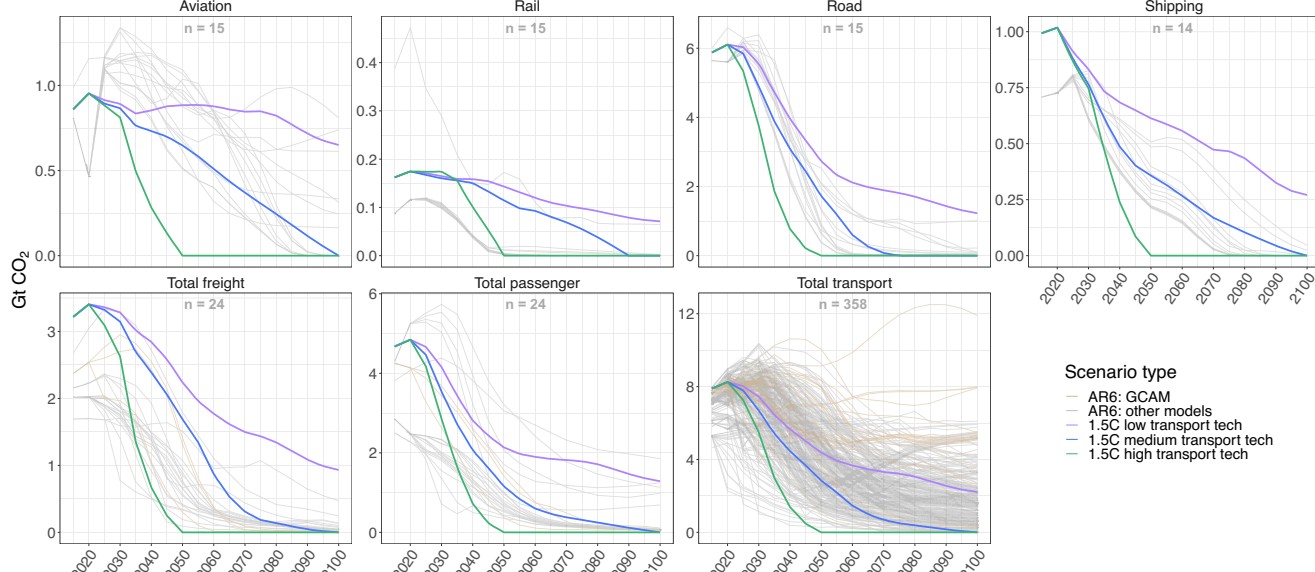

**Fig. 1 | Global transportation CO$_2$ emissions across scenarios. a** Global CO$_2$ emissions from the transportation sector, by mode, across the four scenarios. **b** Cumulative global CO$_2$ emissions from the transportation sector, by mode, from 2020 to 2100. **c** Global CO$_2$ emissions from the transportation sector and from aggregated groups of modes in our three decarbonization scenarios compared to scenarios in the Sixth Assessment Report (AR6) database that are consistent with limiting end-of-century warming to 1.5 °C (with scenarios in the database generated by the Global Change Analysis Model (GCAM) shown in brown and scenarios

generated by all other models shown in gray). The number of scenarios from the database with results for each subset of transportation modes is indicated by the number in gray at the top of each panel. Note that the GCAM scenarios in the AR6 database employed earlier versions of GCAM than the version used in this study, and thus the scenarios may differ in historical periods due to differences in calibration data and other model updates. For a comparable figure showing fuel use by the transportation sector, see Supplementary Fig. 21. HSR high-speed rail.

We supplement GCAM's existing low carbon technology options for the transportation sector (hydrogen, electricity, and biofuels) by adding e-fuels, or synthetic hydrocarbon liquids produced from carbon captured from the atmosphere (see Methods for details). While we consider a full phase-out of liquid fuels in road and rail transportation in our medium and high scenarios, for aviation and shipping we allow liquid fuel use to continue but mandate that this fuel derive from biomass or atmospheric carbon origins rather than fossil fuels (Table 1). A variety of biofuel production technologies exist, with multiple options modeled in GCAM; we specifically require biofuels used in aviation and shipping to be produced via Fischer-Tropsch (FT) synthesis using cellulosic biomass feedstocks. This design choice was made in light of sustainability concerns associated with the direct use of food crops to produce fuel (e.g., corn ethanol, soybean biodiesel) and the suitability of the resulting fuel for use in aviation and shipping[25,36,48–50]. We also impose a constraint on global bioenergy use of 100 EJ by 2100 to address additional environmental concerns related to land competition between biomass cultivation, food

production, and natural lands[51–53], and to attempt to direct biomass consumption towards the highest-value use cases.

## Results

### Consequences for transportation emissions

All decarbonization scenarios achieve deep economy-wide emissions reductions, bringing global mean warming below 1.5 °C by the end of the century (Supplementary Fig. 3). However, the contribution of the transportation sector to these emissions reductions varies between the scenarios (Fig. 1a, b). In the reference scenario, direct emissions from the transportation sector increase by more than 40% from 2020 to 2100. In contrast, direct emissions are eliminated by 2050 in the high transportation technology scenario, with total emissions (direct and indirect) reduced by 99% relative to 2020 (Supplementary Fig. 4). The medium and low transportation technology scenarios maintain residual emissions from transport after 2050, with totals of 39 Gt CO$_2$ and 157 Gt CO$_2$, respectively, of cumulative residual emissions from 2050 to 2100. If these residual emissions were counterbalanced with

direct air capture (DAC) to achieve economy-wide net zero goals, associated costs would be on the order of $7-14 trillion in the medium scenario and $27-54 trillion in the low scenario (see Supplementary Table 1 for DAC cost assumptions). Over the course of the century, the high scenario lowers sector-wide direct emissions by 675 Gt $CO_2$ relative to the reference scenario, and by 245 Gt $CO_2$ relative to the low scenario (Fig. 1b, Supplementary Table 2); the latter value represents about 60% of the carbon budget consistent with a two-thirds chance of limiting warming to 1.5 °C according to the IPCC AR6 report[54]. Non-$CO_2$ and air pollutant emissions also decline correspondingly across our scenarios, with particularly large reductions in CO and NOx emissions (Supplementary Table 3).

By mode, freight trucks and passenger cars and trucks have the largest absolute emissions savings in the high scenario relative to the low scenario over the course of the century (78 Gt $CO_2$ and 70 Gt $CO_2$, respectively), while long-haul and short-haul aviation see the largest percent reductions in cumulative emissions (77% and 75%, respectively). Together, aviation and shipping reduce their emissions by 83 Gt $CO_2$ over the century in the high scenario compared to the low scenario, contributing 34% of the total emissions savings from the transportation sector.

Figure 1c shows transportation $CO_2$ emissions in our decarbonization scenarios compared to scenarios in the AR6 database that achieve the 1.5 °C temperature goal by 2100[55]. For the full transportation sector, our high and medium transportation technology scenarios are well below the median level of emissions from the AR6 scenarios in both 2050 and 2100. Our low scenario is near the median in 2050 but above it in 2100. While there are more than 300 scenarios that model emissions pathways for the combined transportation sector, only 14 scenarios include a detailed breakdown of these transportation sector emissions into aviation, shipping, rail, and road transport. Relative to those scenarios, our high scenario is more ambitious, most notably in aviation and shipping; the scenarios in the AR6 database do not achieve zero emissions from these modes until 2075 or later.

### Contributions of each low carbon technology

The ambitious reductions in transportation emissions in the high scenario are the result of rapid technological transitions across the sector, which are shown in Fig. 2. In the high scenario, most modes are primarily electrified by 2050 onwards, with the exceptions of international shipping and long-haul aviation. International shipping relies principally on hydrogen; it is the only mode for which a larger fraction of service (i.e., transportation activity, in passenger-kilometers or ton-kilometers) is supplied by hydrogen technologies than electric technologies by 2050. Alternative liquid fuels also play a key role in international shipping, particularly in the first half of the century when hydrogen technologies are still emergent. In long-haul aviation, the high costs of using electric batteries or hydrogen combustion systems on long haul flights—barring significant investments in and advancements of these technologies—prevent both technologies from taking off, leading to a dependence on alternative liquid fuels to meet decarbonization targets. These fuels provide over 88% of long-haul aviation service from 2050 onwards in the high scenario. Though these alternative fuels can consist of either biofuels or e-fuels, we find that e-fuels are far less cost-competitive than biofuels (Supplementary Fig. 5) and thus comprise a relatively low fraction of the alternative liquid fuel mix for aviation and shipping (<16% across all years in the primary scenarios). Unlike its long-haul counterpart, short-haul aviation employs a more balanced portfolio of technologies, with electricity providing 56% of service in 2050 and hydrogen and alternative liquid fuels contributing almost evenly to the remainder. In domestic shipping, electricity is dominant, supplying over 80% of service from 2050 onwards in the high scenario; the speed with which this transition occurs implies both a high potential for electrification and an elevated risk of short-haul freight vessels becoming stranded assets.

While these rapid shifts in aviation and shipping technologies will be costly, we find that they could—for most modes—be economically viable in the context of ambitious mitigation goals. By 2050, estimated break-even carbon prices for electric and hydrogen-based technologies (i.e., the carbon prices that would make them cost-competitive with traditional fossil-based technologies) for shipping and short-haul aviation range from 135 to 496 2020$ per t$CO_2$ (Supplementary Table 4), comparable to projected DAC costs of 172−351 2020$ per t$CO_2$ in that year (Supplementary Table 1). Electrifying domestic ships, and employing hydrogen fuel cells on international ones, stand out as becoming cost-competitive at a carbon price of less than 250 2020$ per t$CO_2$. Break-even carbon prices for electric and hydrogen-based long-haul aviation are high (>1300 2020$ per t$CO_2$), further indicative of the financial support and/or substantial technological development required for viability.

Considering other modes, passenger cars and trucks, freight trucks, and passenger and freight rail exhibit high levels of electrification, with hydrogen providing the remainder of service after 2050. Buses rely more substantially on hydrogen than the other modes of road transport, consistent with previous findings[56], with a split of about 44% hydrogen-based service and 56% electricity-based service from 2050 onwards. Note that the steep declines in bus and passenger rail service observed in Fig. 2 occur due to the income-driven shift towards faster modes with lower wait times as global per capita GDP increases; the elevated preference for public transit implemented in our decarbonization scenarios acts against this trend, resulting in a higher share of total vehicle-kilometers for passenger transport provided by buses and rail than in the reference scenario (Supplementary Fig. 6), but does not fully counteract it.

### Differences in technology mix and service across scenarios

Relative to the high scenario, the low and medium scenarios are less ambitious in their scale-up of emerging technologies. Figure 3 compares the service provided by each technology in the three decarbonization scenarios and the reference scenario for aviation and shipping. Other modes and totals for the full transportation sector are shown in Supplementary Figs. 7 and 8. By design, the low scenario has much more remaining oil-based service after 2050 than the medium and high scenarios, particularly in international shipping and long-haul aviation. Biofuels play a more prominent role in the medium scenario than in the high scenario, especially for aviation, due to the less aggressive deployment of hydrogen and electric technologies. Supplementary Fig. 9 shows how these differences in technology mix translate into fuel use across the transportation sector; by 2030, non-fossil fuels comprise 19% of transportation fuel use in the high scenario, more than double their fraction in the same year in the low scenario.

In comparison to the reference scenario, service from all modes is substantially reduced in all three decarbonization scenarios. For most modes, much of this reduction in service is due to the assumption of lower demand for transportation services in the decarbonization scenarios (Supplementary Tables 5 and 6). However, for aviation and shipping, the imposition of the carbon policy further suppresses service due to price elasticity effects. In the low transportation technology scenario, cumulative service provided by aviation and shipping from 2020 to 2100 drops by 40% or more for each mode relative to the reference scenario, with carbon price effects responsible for over half of this decline for international and domestic shipping and short-haul aviation. In that scenario, the difficulty of fuel switching leads to elevated carbon prices (Supplementary Fig. 10) and a reliance on demand destruction to meet emissions mitigation targets. However, ambitious deployment of advanced technologies enables higher rates of fuel switching and partially mitigates these reductions in service, as service

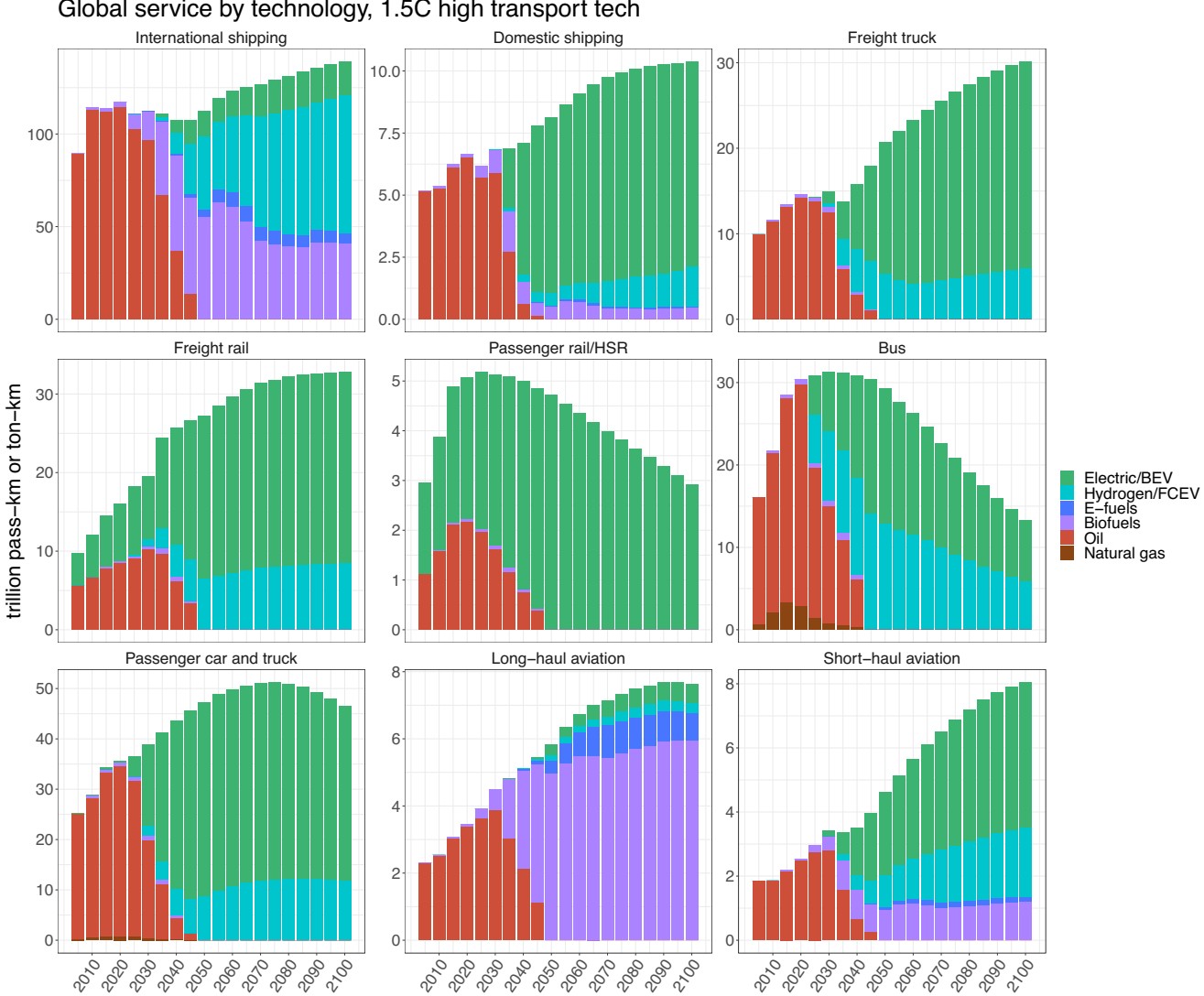

**Fig. 2 | Global service provided by technology for each transportation mode in the 1.5 °C high transportation technology scenario.** Note the different y-axis scales and units for each mode; units are trillion passenger-kilometers (pass-km) for passenger modes (passenger rail and high-speed rail (HSR), bus, passenger car and truck, long-haul aviation, and short-haul aviation) and trillion ton-kilometers (ton-km) for freight modes (international shipping, domestic shipping, freight truck, and freight rail). For aviation, hydrogen technologies employ hydrogen combustion turbines; for all other modes, hydrogen fuel cell electric vehicles are modeled. Also note that there is a very small amount of coal-based freight rail service that is not shown in the figure as it phases out by 2025 and provides only a miniscule contribution to global freight rail transport in the preceding periods (0.009 trillion ton-km of service in 2005, 0.007 trillion ton-km in 2010, and 0.002 trillion ton-km in 2015 and 2020). BEV battery electric vehicle, FCEV fuel cell electric vehicle.

provided by aviation and shipping is higher in the medium and high scenarios than in the low scenario (on the order of 2–7% increases in cumulative service over the century for aviation and international shipping; see further discussion of these and other modes in the Supplementary Notes: Additional Discussion of Shifting Patterns of Transportation Service Across Scenarios). The shifting patterns of transport service across scenarios highlight one of the tradeoffs between different decarbonization pathways that emerge in our low-medium-high scenario framework.

**Impacts beyond the transportation sector**

The differing levels of transportation decarbonization affect the intensity of mitigation required in other sectors (Fig. 4). In the low transportation technology scenario, the reduced level of ambition in the transportation sector forces other sectors to substantially limit their emissions to maintain the same economy-wide emissions constraint. Notably, industry reaches zero emissions by 2100 in that scenario (Fig. 4a, c). The medium and high scenarios allow for less

demanding emissions mitigation outside of the transportation sector, particularly in industry and buildings. Cumulative emissions differences between the high and low scenarios across the century are largest for industry (128 Gt $CO_2$) and electricity (74 Gt $CO_2$) (Fig. 4b, Supplementary Tables 7 and 8). The additional flexibility in emissions mitigation strategies also allows for increased output from industrial sectors in the high transport ambition scenario. Similarly, less carbon capture and storage (CCS) is required in industry in the high transport technology scenario relative to in the low scenario, particularly in the first half of the century (Fig. 4d). Carbon sequestration via bioenergy with carbon capture and storage (BECCS) for both hydrogen and electricity production is also reduced in both the medium and high scenarios as compared to in the low scenario, though generation using fossil fuels paired with CCS increases, likely due to the increased emissions mitigation and elevated use of biofuels in transport. Across the economy, the high transport technology scenario decreases total carbon sequestration by 0.2–0.7 Gt $CO_2$ relative to the low scenario in all periods from 2030 onwards.

## Global service by technology for aviation and shipping

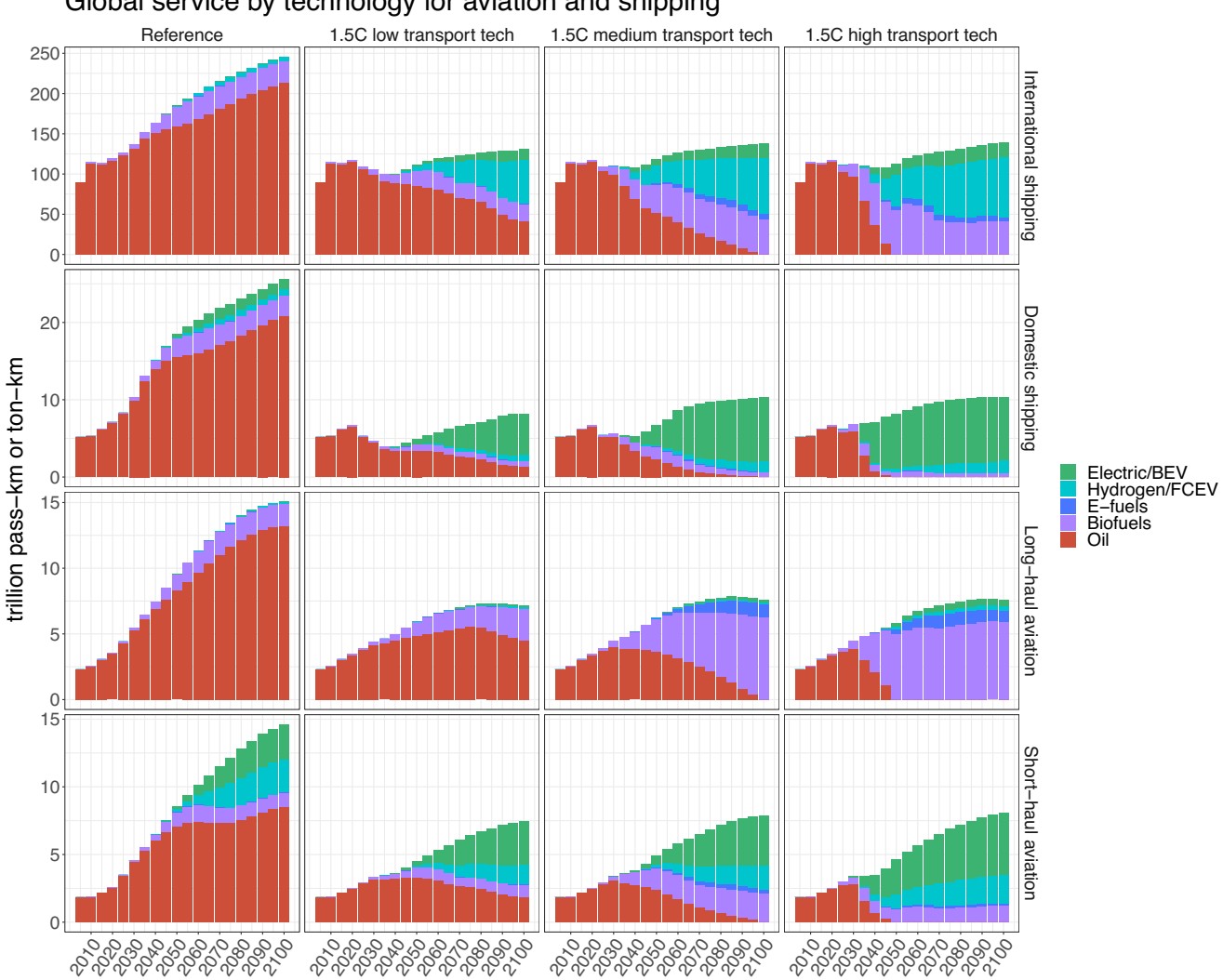

**Fig. 3 | Global service provided by technology for international and domestic shipping and long-haul and short-haul aviation across the four scenarios.** Note the different y-axis scales and units for each mode; units are trillion passenger-kilometers (pass-km) for aviation and trillion ton-kilometers (ton-km) for shipping.

For aviation, hydrogen technologies employ hydrogen combustion turbines, while for shipping, hydrogen fuel cell electric vessels are modeled. BEV battery electric vehicle, FCEV fuel cell electric vehicle.

Looking beyond emissions, the medium and high scenarios increase demand for both hydrogen and alternative liquid fuels in the transportation sector but lead to reduced use of these fuels in other sectors (Fig. 5a, b). For hydrogen, higher transport demand is driven especially by road vehicles and shipping (Supplementary Fig. 11). In the high scenario, the transportation sector consumes more than 17 EJ of hydrogen by 2050, or 69% of all hydrogen consumption, as compared to 9.4 EJ (47%) in the medium scenario and 5.5 EJ (28%) in the low scenario. The high scenario also increases economy-wide hydrogen demand in 2050 by over 25% relative to the low scenario, though by 2100 the scenarios have nearly equalized. For the alternative liquid fuels, from 2025 onwards, all FT biofuels and e-fuels produced are consumed by aviation and shipping, with aviation using the majority (Supplementary Table 9 and Supplementary Fig. 12). The increased consumption of biofuels and e-fuels by transportation limits their availability for other sectors, as the fraction of biofuels and e-fuels in refined liquids consumed by non-transportation sectors is lower in the medium and high scenarios than in the low scenario from 2030 onwards (Supplementary Table 10). The industrial sector, for example, uses 11.5 EJ of biofuels and e-fuels in 2100 in the low scenario but only 8 EJ in the high scenario (Fig. 5b). Similar shifts occur between the

industrial and transport sectors in terms of their electricity consumption; in 2050, the transport sector consumes 14 EJ more electricity in the high scenario relative to in the low scenario (39 EJ vs 25 EJ, respectively), while the industrial sector consumes 20 EJ less (Supplementary Fig. 13).

The different levels of ambition in the transport sector and their associated fuel demands also have consequences for the upstream generation of those fuels (Fig. 5c, d). In all three decarbonization scenarios, 99% of electricity and 98% or more of hydrogen is produced using low carbon technologies by 2050. The high ambition transport scenario requires increased electricity generation in early periods relative to the low ambition scenario due to the rapid electrification of transportation modes, with most of this increased generation initially supplied by fossil sources while renewable sources and CCS technologies are still scaling up (Fig. 5c). After mid-century, both the medium and high scenarios see less electricity production than the low scenario, particularly via BECCS and solar power. In hydrogen production, most of the elevated hydrogen demand in the high transport technology scenario is met by increased generation via electrolysis using grid electricity (Fig. 5d). Hydrogen production using BECCS is notably reduced in the higher

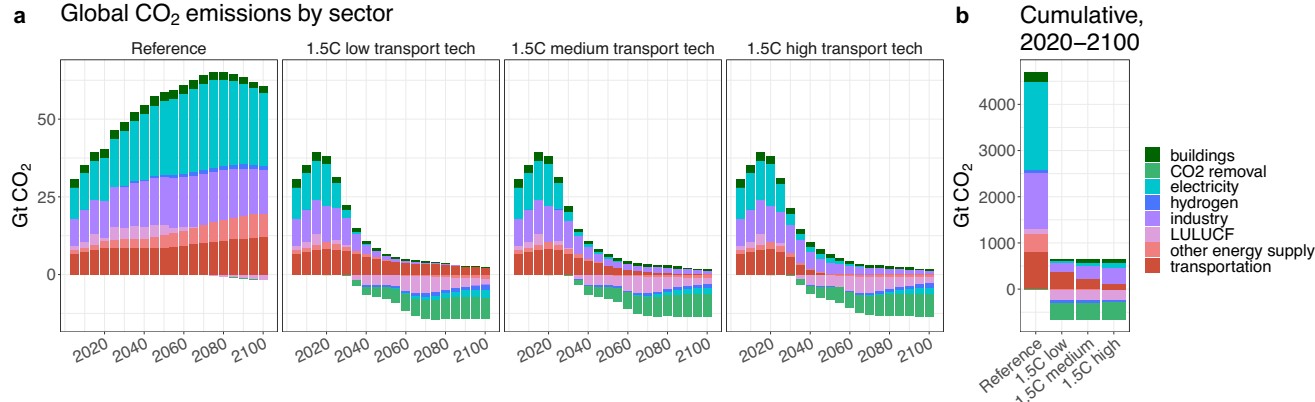

**a** Global CO$_2$ emissions by sector

**b** Cumulative, 2020–2100

**c** Global CO$_2$ emissions from selected sectors

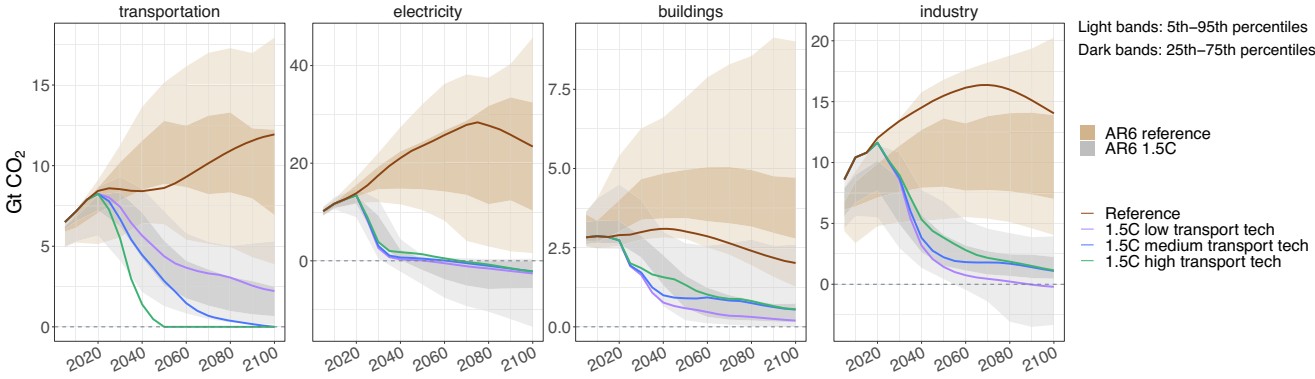

**d** Global CO$_2$ sequestration

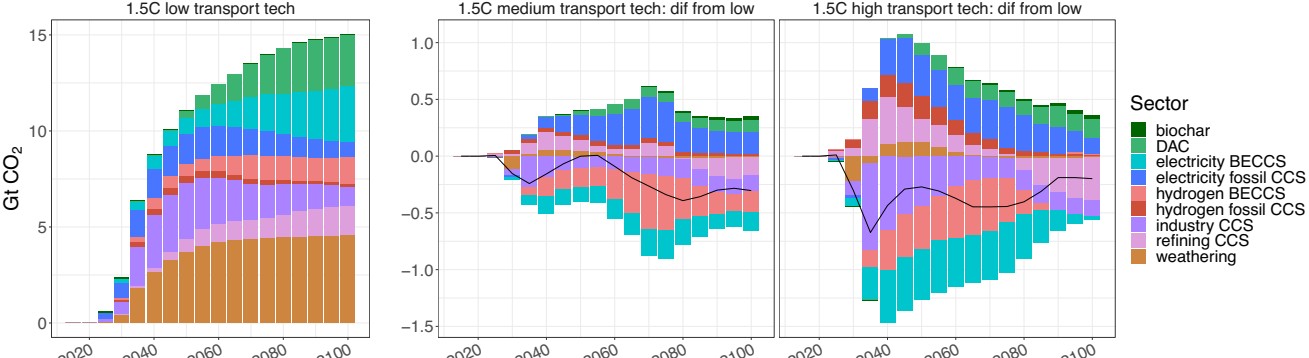

**Fig. 4 | Global total CO$_2$ emissions and sequestration across scenarios. a** Global CO$_2$ emissions by sector across the four scenarios. **b** Cumulative global CO$_2$ emissions by sector, from 2020 to 2100, for each scenario. **c** Comparison of global CO$_2$ emissions from transportation, electricity, buildings, and industry in the four scenarios and relative to the scenarios in the IPCC Sixth Assessment Report (AR6) database that are consistent with limiting end-of-century warming to 1.5 °C (gray bands) and that represent a continuation of baseline trajectories (brown bands). The bands indicate the 5th to 95th percentiles (light) and the 25th to 75th percentiles (dark) from the AR6 scenarios. Note the different y-axis scales for each sector. Also note that GCAM includes a detailed industrial sector breakdown that incorporates agricultural energy use, aluminum, cement, chemicals, construction, iron and steel, mining, fertilizer, industrial processes, and other non-specified industry; other integrated assessment models may differ in their definition and scope of the industrial sector. **d** Global CO$_2$ sequestration by sector across the scenarios. Sequestration from industrial feedstocks is not included in the figure. Note that results for the 1.5 °C medium and high transport technology scenarios are shown as a difference relative to the 1.5 °C low transport technology scenario, with the black line indicating the net difference between the scenarios and a positive value indicating more sequestration in the medium/high scenario than in the low scenario. LULUCF land use, land use change, and forestry, DAC direct air capture, BECCS bioenergy with carbon capture and storage, CCS carbon capture and storage.

ambition transport scenarios, contributing only 5% (1.3EJ) of hydrogen generation in the high scenario in 2050 as compared to 17% (3.4 EJ) in the low scenario.

**Sensitivity scenarios**

We primarily base our analysis on SSP1 assumptions, which are consistent with sustainable, low emissions socioeconomic pathways, and transportation demand reduction assumptions that simulate a societal willingness to limit the movement of people and goods to help meet climate goals. However, to evaluate the robustness of our findings and the identified transition pathways, we also assess the feasibility of implementing the high ambition technological transition in the transportation sector and meeting the same decarbonization targets under less optimistic assumptions (Fig. 6). We consider seven

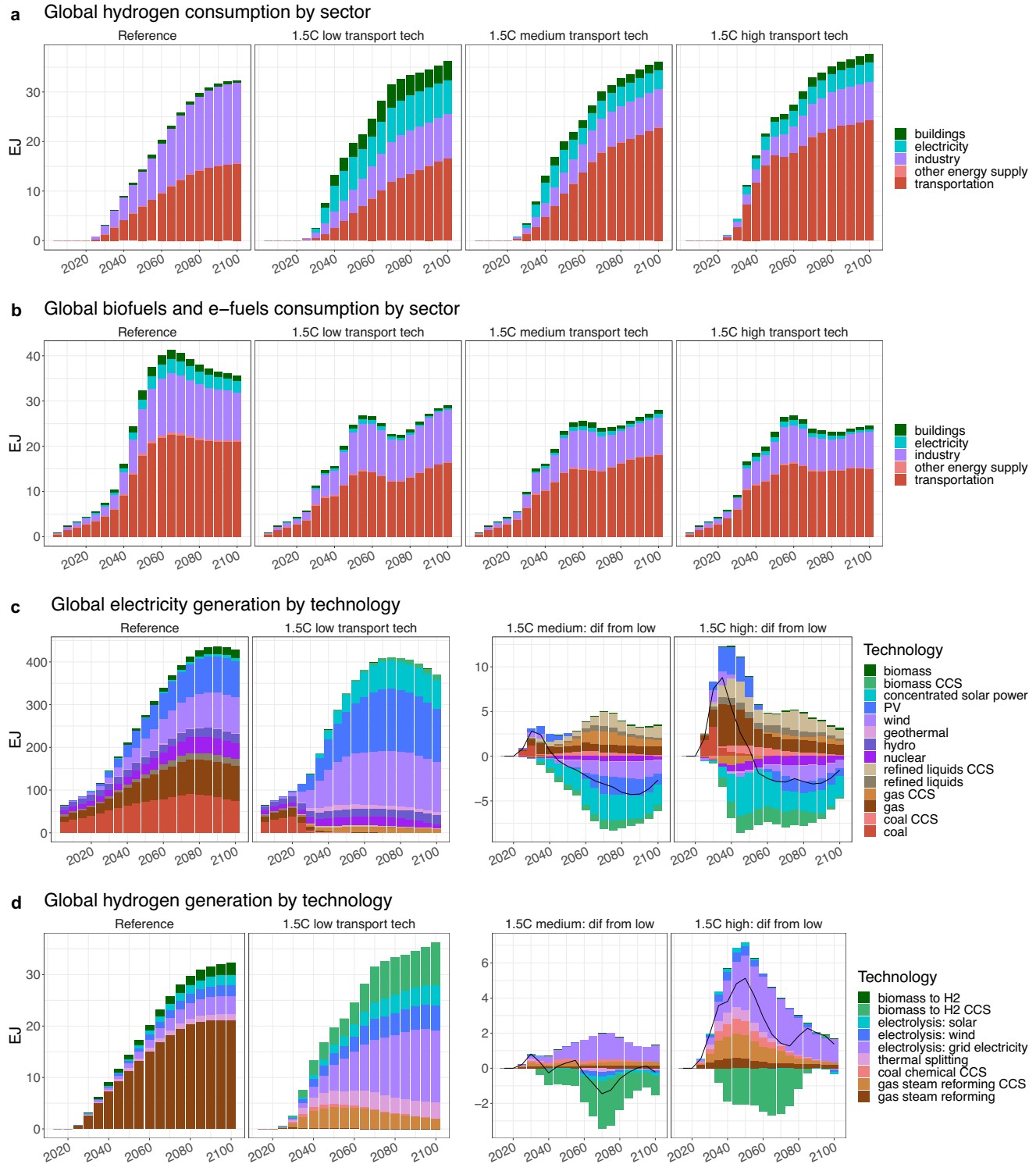

**Fig. 5 | Global fuel consumption and generation across scenarios for selected fuels. a** Global hydrogen consumption by sector across the four scenarios. **b** Global aggregated biofuels and e-fuels consumption by sector across the four scenarios. **c** Electricity generation by technology across the four scenarios. **d** Hydrogen generation by technology across the four scenarios. Electrolysis technologies are listed by the source of the electricity used for electrolysis: either generated using wind or solar power, or supplied by the grid. Note that in **c** and **d**, results for the 1.5 °C medium and high transport technology scenarios are shown as a difference relative to the 1.5 °C low transport technology scenario, with the black line indicating the net difference between the scenarios and a positive value indicating more generation in the medium/high scenario than in the low scenario. CCS carbon capture and storage, PV photovoltaics, H2 hydrogen.

sensitivity cases. The first four employ population and GDP projections consistent with each of the alternative SSPs[45] (Supplementary Fig. 14); though storylines vary between the SSPs, SSPs 2–4 generally increase inequality in growth and development relative to SSP1, while SSP5 represents a rapid economic growth pathway fueled by high levels of

energy use. We also consider two sensitivity scenarios related specifically to transport demand: one with standard—rather than reduced—transportation service demand as a function of income, using base GCAM values for transportation income elasticity, and one with a decrease in the responsiveness of transportation demand to shifts in

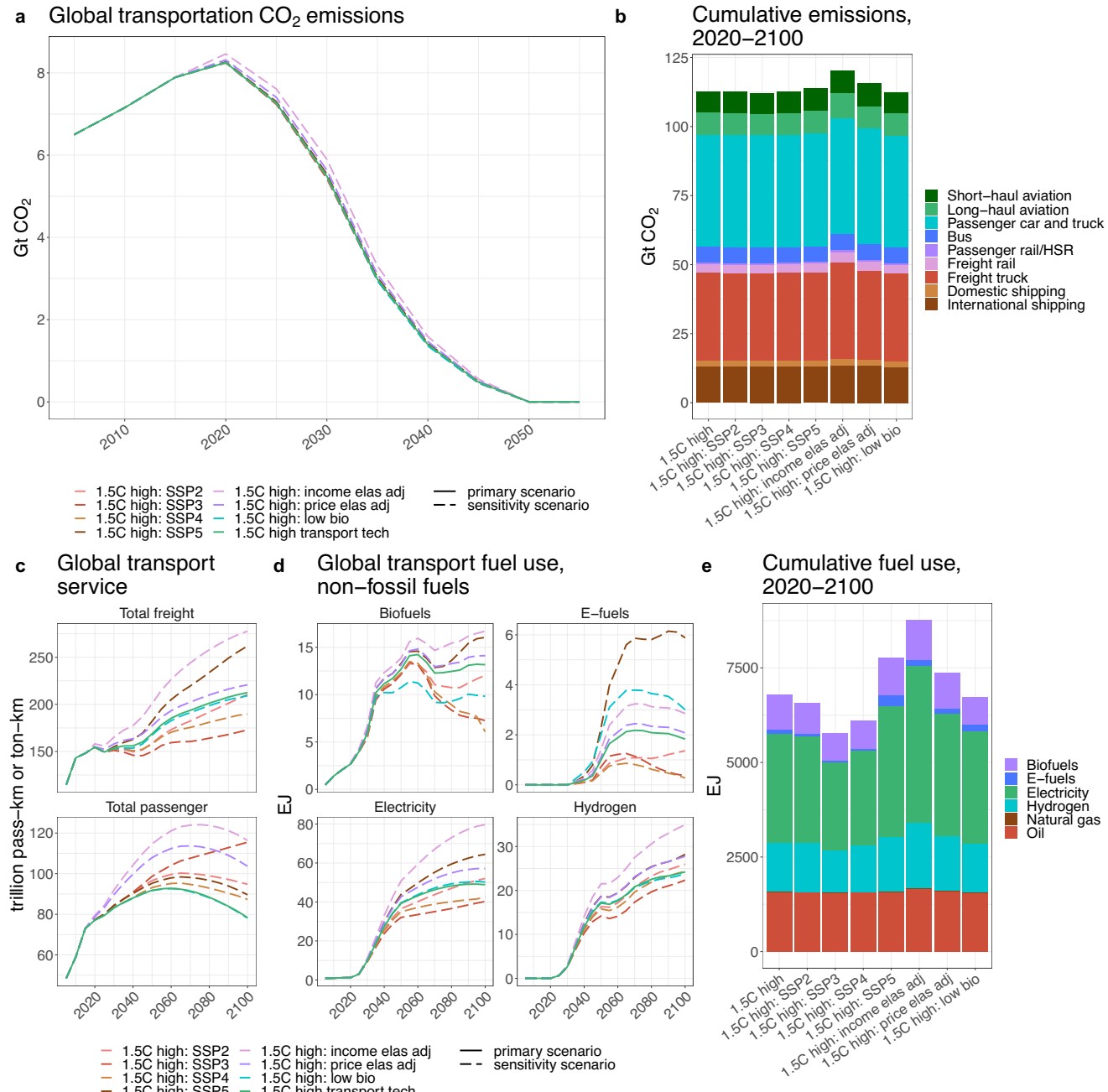

**Fig. 6 | Sensitivity analysis.** Global results for the high transportation technology scenario and the sensitivity scenarios for **a** transportation CO$_2$ emissions, **b** cumulative transportation CO$_2$ emissions by mode from 2020 to 2100, **c** transportation service by passenger and freight modes, **d** non-fossil fuel use by the transportation sector, and **e** cumulative fuel use by the transportation sector from 2020 to 2100. In **a**, **c**, and **d**, the high scenario is shown with a solid line and the sensitivity scenarios are shown with dashed lines. Note that biofuel use spikes starting in 2050 due to the imposition of the fossil phase-out in transport in 2050.

In **e**, there is a very small amount of coal use in transport, exclusively utilized by freight rail prior to 2025, which is not shown in the figure as it provides only a miniscule contribution to global transport fuel use (0.007 EJ cumulatively over 2020–2100 in all scenarios). HSR high-speed rail, pass-km passenger-kilometers, ton-km ton-kilometers, SSP Shared Socioeconomic Pathway, income elas adj = income elasticity adjustment, price elas adj price elasticity adjustment, low bio low bioenergy.

prices (i.e., reduced price elasticity), limiting the decline in transport service due to carbon price effects. Finally, we include a sensitivity case in which economy-wide bioenergy use is subject to a stricter constraint, maximizing at 70 EJ in 2100, than in our standard 1.5 °C-consistent scenarios.

While all sensitivity scenarios mitigate emissions from the transportation sector comparably to the standard high ambition scenario, the fuel and resource demands required to realize this mitigation vary widely. Cumulative emissions differences from 2020 to 2100 between

the sensitivity scenarios and the standard high transportation technology scenario are less than 8 Gt CO$_2$ (<7% of total cumulative emissions) (Fig. 6a, b). Achieving commensurate levels of emissions reductions in the income and price elasticity adjustment scenarios, which increase transport demand (Fig. 6c), requires elevated use of low carbon fuels (Fig. 6d). Combined hydrogen and electricity consumption by the transportation sector in 2100 is more than 50% higher in the income elasticity adjustment scenario than in the standard high ambition scenario, and more than 15% higher in the price elasticity

adjustment scenario. In the SSP2, SSP3, and SSP4 scenarios, passenger transport service demand is higher than in the standard high scenario primarily due to the larger global population in those scenarios, while freight service demand is lower due to reduced global GDP (Fig. 6c and Supplementary Fig. 14). In the SSP3 and SSP4 scenarios, the net effect is an overall reduction in energy demands for transport across fuel types, though to a lesser degree for hydrogen than for other fuels (Fig. 6d, e); the elevated carbon prices in these scenarios (over 50% higher than in the standard high scenario by late-century; see Supplementary Fig. 15) increase the cost-competitiveness of hydrogen, particularly for international shipping. In the SSP2 scenario, total energy use by transport is similar to the standard high scenario, but the fuel breakdown shifts: biofuels are used less, replaced by hydrogen and, in later years, electricity (Fig. 6d, e). This likely results in part from a reduction in purpose-grown biomass production in the SSP2 scenario due to land use competition with crop production, as food demand and associated land requirements increase with the larger SSP2 population (Supplementary Fig. 16). The SSP5 scenario sees higher transportation demand and elevated use of all fuels, but the increase in e-fuel use (more than triple in the standard high scenario by 2100) is especially notable. E-fuels, particularly those produced via on-site hydrogen electrolysis powered by wind energy, achieve their lowest costs in this scenario (Supplementary Fig. 17). In the more constrained bioenergy sensitivity case, transport biofuel consumption is 25% lower in 2100 than in the standard high scenario, corresponding to a 3 EJ difference which is mostly compensated for by elevated electricity and e-fuel consumption (Fig. 6d, e). Aside from the impacts of these differences in biofuel and e-fuel use, the relative contributions of the advanced technologies to the decarbonization of each transport mode are largely robust across the standard high scenario and the sensitivity scenarios (Supplementary Figs. 18 and 19; see further discussion in the Supplementary Notes: Additional Discussion of Sensitivity Scenarios).

## Discussion

Aggressive technological development and an earlier fossil phase-out in the transportation sector can substantially reduce carbon emissions from transport, complementing the emissions cuts required in other sectors. In our high transportation technology scenario, the transportation sector achieves zero direct carbon emissions by 2050. In contrast, in the medium and low scenarios, residual transportation emissions after 2050 must be offset by negative emissions in other sectors or $CO_2$ removal technologies to achieve economy-wide net zero emissions. We find particularly notable interactions between the transportation and industrial sectors, in that when transportation decarbonization ambition is low, ambition in industry must be high to compensate, and when transportation ambition is high, industry has more flexibility in emissions reductions strategies. We also observe lower deployment of BECCS for both hydrogen and electricity production in our high ambition transport scenario, likely due to both elevated use of biofuels by the transportation sector (in the context of the economy-wide constraint on bioenergy) and the reduced need for carbon sequestration in that scenario due to increased mitigation by transport. Within the transportation sector, higher ambition also partially alleviates the reductions in aviation and shipping service due to the stringent emissions constraint, allowing for more transport service without compromising climate goals. Exploration of these inter- and intra-sectoral dynamics is important when examining pathways for achieving zero emissions, particularly when considering deep decarbonization across multiple sectors and the resulting implications for alternative fuel demands and technology shifts.

Achieving such ambitious levels of transportation decarbonization will require technological transitions across the sector. We find that electrification contributes most to this decarbonization, particularly for road and rail transport, consistent with other studies[4,13,15,23,30,32]. Hydrogen and biofuels are also crucial for long-distance travel in

aviation and shipping. While we observe that the decarbonization of long-haul aviation heavily depends on biofuel use, synthesis of biofuels via the FT pathway will still require advanced research, development, and commercialization to be used on the scale necessary to satisfy aviation biofuel demand[48,57]. Given these challenges, as well as potential limitations on bioenergy use and the increasing investments in the development of hydrogen propulsion systems for aviation by some industry leaders[58], it is possible that hydrogen-based technologies could outpace biofuels to provide a larger share of long-haul aviation service than indicated by our results. In our primary scenarios, e-fuels only see limited utilization due to their relatively high costs, supporting prior analyses[21,49,59]. However, we also observe that under high economic growth (SSP5) assumptions and/or tightly constrained economy-wide bioenergy availability, e-fuel deployment increases, particularly for long-haul aviation. These results offer insights into the conditions under which some industry pathways and other analyses that rely on e-fuels for aviation decarbonization may be more likely to be realized[13,60]. E-fuel subsidies or blending mandates may further increase their utilization, which could reduce the need for other low carbon transportation fuels but may lead to overall higher costs associated with decarbonizing the sector. Other zero emissions fuels that we do not model, most notably ammonia, are also being considered for aviation and shipping[19,61]; additional integrated modeling could analyze the potential role of these fuels in transportation decarbonization pathways and the resulting costs and benefits, taking into consideration the potential implications of widespread ammonia use on the nitrogen cycle[62].

By identifying key low carbon technologies for each transportation mode, our results can help guide the development and prioritization of emerging technologies, facilitating the transition to a 1.5 °C-aligned future. Such information is crucial given the speed that is required for this transition to be successful: even in the low transportation technology scenario, passenger transport service provided by low carbon technologies triples between 2020 and 2030, and freight service almost doubles. In the high scenario, service from alternative technologies grows by eight times in passenger transport in this ten-year period, and almost three times in freight.

Realizing this rapid scale-up of emerging technologies would pose a variety of challenges for resource and fuel availability, infrastructure development, and supply chain management. As electric and hydrogen-based technologies for many modes, particularly aviation and shipping, are still nascent, their utilization at the scale observed in our high ambition scenario would depend on continued advancement and successful market entry[2,12,13]. High levels of transport electrification would necessitate the widespread deployment of charging infrastructure, for vehicles ranging from passenger cars to large cargo ships, and produce a large demand for critical minerals for battery development[10,31]. Similarly, ubiquitous transportation hydrogen use would require the large-scale development of vehicle technologies, as well as hydrogen transmission and distribution networks and/or on-site electrolysis systems at refueling stations[10,49]. To ensure that electric and hydrogen-based vehicles are truly zero emissions beyond the tailpipe, the upstream production of these fuels must also decarbonize[10,63,64]; the transition from current fossil fuel-dominated production would need to occur alongside an expansion in generating capacity to meet rising electricity and hydrogen demand[13]. The production of biofuels for use by the transportation sector would also need to scale up by almost 5 times relative to 2020 levels and shift primarily to advanced generation technologies to meet aviation and shipping biofuel demands of more than 13 EJ by 2050 in the high ambition scenario. This biofuel production would have resulting consequences for land use and associated emissions that will depend on the specific biofuel generation pathways that are most widely employed[49,63,65–67]. Emerging technologies that co-produce multiple forms of biofuels intended for use in different transportation modes or

other applications could also help address some of the tradeoffs in biofuel usage that arise in our scenarios[25]. For all low carbon technologies, ensuring that sufficient financial capital and human resources are available at all stages of the supply chain—including procurement of raw materials and resources, technology manufacturing and deployment, and infrastructure siting and development—will be critical to avoid any bottlenecks that could constrain or impede the massive scale-up required[31,68].

Our results suggest that financial and policy incentives, including mechanisms such as carbon prices, could be crucial to ensure the competitiveness of low carbon technologies in transport. Future research evaluating specific policies and incentive structures, as well as quantifying the associated resource and development needs at each stage of the supply chain, would complement our work and facilitate the development of sectoral decarbonization plans. In our analysis, we maintain the same assumptions for demand reduction, transit and ride-sharing preference, and efficiency improvements in all our decarbonization scenarios. Building on both our work and other studies[31,34,69] by considering alternative scenarios for these behavioral changes and other consumer preferences could add another valuable dimension to modeling of pathways for transport decarbonization. As we observe substantial reductions in transport non-$CO_2$ emissions in our high ambition scenario, additional studies could quantify the benefits for air quality and health of an earlier elimination of fossil fuel use in the sector[70]. Incorporating those benefits into an integrated assessment of the distributional impacts of transportation decarbonization strategies would complement our findings, enabling a robust consideration of economic, environmental, and equity implications of technological transitions in the transportation sector.

## Methods

We use GCAM version 6.0, with all scenarios run on the Pacific Northwest National Laboratory's high performance computing cluster, deception. GCAM's assumptions for conventional, fossil fuel-based transportation technologies are documented in Mishra et al.[71]. Assumptions for advanced, low carbon transportation technologies have been updated with recent model development and are available in the GCAM documentation[72]. As maritime shipping and aviation are a focus of our work, we also include information on the non-fuel costs and energy intensities assumed for these modes in the Supplementary Methods (Shipping and Aviation Technology Assumptions section), particularly detailing the assumptions employed for shipping and aviation low carbon technologies. Aside from the scenario assumptions and new modeling capacity discussed below, all other technological and socioeconomic parameters match those in the core release of GCAM version 6.0[73].

GCAM includes a hybrid, or high efficiency, technology option for refined liquids-based transportation technologies, facilitating the endogenous representation of efficiency improvements. As this is not a focus of our work, however, in our results we aggregate the high efficiency liquids technologies with their corresponding standard efficiency counterparts when showing transportation service by technology. GCAM also includes walking and cycling modes, which we do not feature in our results as they contribute relatively small shares of total passenger transport and do not demand any energy.

We added the capability to model synthetic hydrocarbon fuels using carbon captured from the atmosphere via direct air capture (DAC) and hydrogen as prospective drop-in replacements for today's petroleum-derived liquid fuels. We parameterized four technologies for synthetic fuels production, which differ in how both the required electricity and hydrogen are produced. This is intended to represent varying degrees of stringency regarding the use of fossil fuels to generate the electricity and/or produce the hydrogen and allow each of these technologies to compete on a cost basis. The first technology uses grid electricity to capture $CO_2$ from the atmosphere and

purchases an industrial hydrogen commodity which may either be produced on-site or delivered via pipeline or liquid truck from a centralized hydrogen production facility. The second technology uses grid electricity to capture $CO_2$ from the atmosphere and electrolyze hydrogen on-site. The last two technologies use electricity generated by dedicated wind turbines or solar panels, respectively, to run both the DAC and hydrogen electrolysis processes and are intended to represent the most restrictive definition for zero-carbon hydrogen and electricity sourcing. The levelized capital and fixed operating costs for the hydrogen electrolyzers and DAC equipment for these technologies are adjusted using regionally explicit capacity factors for wind turbines and solar panels and are harmonized with GCAM's electricity sector[74,75]. Parametric assumptions for hydrogen production and distribution are provided in the GCAM 6.0 release[73]. The DAC cost and energy performance parameters assume a high-temperature, fully-electric liquid solvent-based process most similar to the one being developed by Carbon Engineering, as this is, to our knowledge, the most detailed publicly-available cost and performance data for a commercial DAC to liquid fuels process[76]. The derivation of GCAM input assumptions for this process is documented in Fuhrman et al.[77]. The energy and non-fuel cost coefficients for DAC are multiplied by 19.6 kg C per GJ of fuel, consistent with GCAM's existing refined liquids commodity, and 1.19 GJ of hydrogen is assumed to be required per GJ of liquid fuels produced[78]. Efforts to evaluate alternative sources of captured $CO_2$, including waste $CO_2$ and low-temperature DAC processes in GCAM are the subject of separate studies. We do not consider ammonia as a potential zero carbon fuel due to the large uncertainty regarding fugitive emissions that could further disrupt planetary boundaries for reactive nitrogen and fully negate any climate benefit achieved by avoiding $CO_2$ emissions[62]. Efforts to model DAC-to-methanol as a means of decarbonizing the petrochemical sector are also left as an area for future work, as this technology is not considered in this study.

In our cost calculations, we use the DAC costs shown in Supplementary Table 1 to calculate costs that would be required to offset residual post-2050 transportation emissions with DAC. These costs are derived from GCAM assumptions for DAC non-energy costs and energy coefficients, as well as endogenously calculated GCAM energy costs, and are linearly interpolated between the years shown in Supplementary Table 1. In calculating the break-even carbon prices for electric and hydrogen-based shipping and aviation technologies (Supplementary Table 4), we compute the difference in costs (on a per service output basis) between the low carbon technologies and the corresponding standard refined liquids-based technology, divided by their difference in emissions per service output. We only consider direct emissions in this calculation; thus, electric and hydrogen-based technologies are assumed to generate zero emissions. Cost values are used from the reference scenario and include both energy and non-energy costs.

### Emissions constraint

In our 1.5 °C scenarios, we impose a carbon emissions constraint that limits global warming to below 1.5 °C in 2100. This constraint applies to fossil fuel and industry carbon emissions and begins in 2025, GCAM's first timestep beyond the present. Carbon emissions from land use are excluded from the constraint but are priced at an increasing fraction of the carbon price applied to fossil emissions; this encourages carbon storage through land use change, but only as an addition to the mandated emissions reductions from fossil fuel and industry sources. Non-$CO_2$ greenhouse gas emissions are also not constrained, but their forcing impacts are incorporated in GCAM's temperature projections.

For comparisons with scenarios in the IPCC AR6 database, we select 1.5 °C scenarios using their FaIRv1.6.2 category (C1a, C1b, and C2: below 1.5 °C with no, low, or high overshoot, respectively). We select reference scenarios using their policy category name (P1a: Baseline). In

**Table 2 | Adjustments to income elasticities in the 1.5 °C high, medium, and low transportation technology scenarios**

| Modes | Default value | Adjusted value | Notes |
|---|---|---|---|
| Long-haul aviation | 1 | 0.75 | 25% reduction |
| All other passenger | 1 | 0.8 | Using SSP1 value provided in GCAM |
| International shipping | 0.4 | 0.3 | 25% reduction |
| All other freight | 0.75 | 0.5 | Using SSP1 value provided in GCAM |

*GCAM* Global Change Analysis Model, *SSP1* Shared Socioeconomic Pathway 1.

**Table 3 | Detailed description of the sectoral strengthening scenario**

| Element | Assumptions from Fuhrman et al. | This study |
|---|---|---|
| Population + Socioeconomics | Population peaks at approximately 8.5 billion by mid-century and returns to 7 billion by 2100, following SSP1 | Same |
| Industry | Improved material and energy efficiency; reduced demand | Same |
| Buildings | Higher energy efficiency; lower residential floorspace satiation values | Same |
| Agriculture and Land-use | Reduced preference for beef and non-staple crops | Reduced preference for all animal proteins and non-staple crops |
| Electricity | Faster phase-in of wind and solar; reduced social preference for nuclear | Same |
| Transportation | Increased ride sharing; reduced shipping + aviation demand; phase-outs of internal combustion freight and passenger vehicles excluding aviation + marine shipping and higher preference for electric vehicles | Increased ride sharing and public transportation use; reduced demand for all transportation modes; more aggressive phase-outs and technological changes (see scenario assumptions for details) |
| Bioenergy | Global bioenergy consumption constrained to 56 EJ in 2050 and 100 EJ in 2100, which reduces BECCS | Same |
| Geologic carbon storage | Standard GCAM on-shore geologic carbon storage cost assumptions are increased by 10×; no offshore carbon storage availability | Same |
| Engineered CDR | DACCS and EW are constrained to 8 $GtCO_2\text{-}yr^{-1}$ | Not used |
| Non-$CO_2$ | Reduced $CH_4$ emissions from dairy, rice, and beef through technological improvement; all regions meet Kigali amendment targets for F-gases | Same |

Taken from Fuhrman et al.[44], with modifications as noted to indicate where our 1.5 °C scenarios differ. *GCAM* Global Change Analysis Model, *SSP1* Shared Socioeconomic Pathway 1, *BECCS* bioenergy with carbon capture and storage, *CDR* carbon dioxide removal, *DACCS* direct air capture with carbon storage, *EW* enhanced weathering.

calculating $CO_2$-equivalent emissions from non-$CO_2$ greenhouse gases for our scenarios, we use global warming potentials from the AR5 report; we compare these values to the emissions of Kyoto gases from scenarios in the IPCC AR6 database. We compare the temperature projections for our scenarios generated by Hector, the reduced form carbon-cycle climate model linked to GCAM, to the 50th percentile FaIRv1.6.2 surface temperature variable for the AR6 scenarios.

### Scenario assumptions

To vary the level of technological ambition between our 1.5 °C scenarios, we adjust the rates at which electric and hydrogen-based transportation technologies phase in and fossil-based technologies phase out. For aviation and shipping, the elimination of fossil fuels occurs through the alternative liquid fuel blending mandates shown in Supplementary Fig. 20, representing the minimum fraction of liquid fuels for use in aviation and shipping that must be supplied by e-fuels or FT biofuels in each year. For road and rail transportation, liquid fuel use is completely discontinued by 2100 in the medium ambition scenario and by 2050 in the high ambition scenario. In the low ambition scenario, we impose no blending mandate for aviation and shipping nor a phase-out of fossil fuel use in the transportation sector.

In all the primary 1.5 °C scenarios, we assume lessened demand for transportation services as well as more ridesharing and public transportation use. We implement these assumptions by reducing the income elasticities for transportation (as shown in Table 2), increasing the load factors for 4-wheel light duty vehicles linearly so that a 25% improvement is achieved in 2050, and using the SSP1 assumptions for value of time traveled to encourage use of public transport.

We also implement a variety of assumptions consistent with sustainable development pathways outside of the transportation sector to ensure that our scenario design incorporates comparable levels of increased mitigation across all sectors. These assumptions include reduced population growth, increased energy and material efficiency in the buildings and industrial sectors, reduced demand for industrial goods and services, faster transition to renewable electricity sources, dietary preference changes leading to lessened consumption of animal products and non-staples, constraints on overall bioenergy production, and achievement of the reduction in HFC emissions established by the Kigali Amendment to the Montreal Protocol. Most shifts match those in the 1.5 °C "sectoral strengthening" scenario from Fuhrman et al.[44], as shown in Table 3.

Our sensitivity scenarios are implemented as modifications of our standard 1.5 °C high transportation technology scenario; except as specified in the following, all input parameters match those in the standard 1.5 °C high scenario. In the 1.5 °C high: income elasticity adjustment scenario, we use GCAM's default income elasticity values, shown in Table 2, rather than the reduced income elasticity values employed in all other 1.5 °C-consistent scenarios. In the 1.5 °C high: price elasticity adjustment scenario, we reduce the price elasticity for transportation by 20% from the default value in GCAM. In the SSP sensitivity scenarios, we employ population and GDP trajectories consistent with each of the respective SSPs (Supplementary Fig. 14). SSP3 and SSP5 place high pressure on the global economic, energy, and climate systems due to their rapid growth in population (SSP3) or GDP (SSP5). To ensure the feasibility of achieving the 1.5 °C-consistent emissions constraint in those scenarios, we increase their respective

negative emissions budgets (i.e., the maximum allowed gross value of negative emissions, as a percent of GDP) by 50% above their default values in GCAM. In the low bioenergy sensitivity scenario, we further limit economy-wide bioenergy availability beyond the constraint implemented in the other 1.5 °C-consistent scenarios; bioenergy use maximizes at 70 EJ in 2100 in this sensitivity case, as compared to 100 EJ in the standard 1.5 °C high transport technology scenario.

## Reporting summary

Further information on research design is available in the Nature Portfolio Reporting Summary linked to this article.

## Data availability

GCAM is an open-source community model available at https://github.com/JGCRI/gcam-core/releases. The version of GCAM and other input files used in this work, along with the generated output datasets of GCAM results that support the findings of this study, are available at https://doi.org/10.5281/zenodo.10211171[79].

## Code availability

Scripts used to process and analyze GCAM output data and produce figures are available in the Zenodo repository at https://doi.org/10.5281/zenodo.10211171[79].

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

## Acknowledgements

This research was supported by the ClimateWorks Foundation. HM was also supported by the National Research Foundation of Korea (BP Grant: RS-2023-00219466).

## Author contributions

S.S., J.F., L.A.L., S.M., and H.M. designed the research. S.S. led the modeling and wrote the first draft of the paper. S.S., J.F., M.G., P.K., and H.M. contributed to the modeling tools. S.S., J.F., L.A.L., S.M., and H.M. reviewed and interpreted the results. All authors contributed to writing the paper.

## Competing interests

The authors declare no competing interests.
