## [Peer Review File · Nature Communications]

Integrated assessment modeling of a zero-emissions global transportation sectorReviewers' Comments:

Reviewer #1:

Remarks to the Author:

This paper employs the global integrated assessment model GCAM to evaluate a variety of 1.5C-consistent scenarios for the transportation sector, considering varied timelines for the phase-out of fossil fuels and the implementation of advanced alternative technologies.

On the one hand, the model is top-of-class; the analysis is solid; and the results are interesting. On the other hand, the methodological development here is minimal; the experimental set-up is not particularly creative; and the insights are far from surprising.

The authors state the following on p. 2:

"While there are a variety of industry scenarios and analyses for aviation and shipping decarbonization in isolation, these do not employ an integrated framework that considers the connections between the transportation sector and other sectors and environmental systems."

This is the real contribution that a paper of this type can make to the literature, failing a presentation of novel methodological developments. I would highly suggest that the author focus much more of their paper on systems interactions (both results and insights). The section starting on p. 10 get into this, but does not go far enough.

The quality of written English in the paper is strong, and the presentation and organization are very good.

Some more specific comments below...

Line 43 => The lit review in this and other sections is really excellent.

Line 94 => More information is needed to describe what exactly is an e-fuel in the GCAM context (with this modified version 6.0)? The Methods section contains a few words, but not enough.

Line 99 => There are other biofuel conversion pathways that do not lead to competition with food crops. Why only allow the FT-cellulosic pathway?

Line 166 => In Figure 2, why are passenger rail and bus declining so quickly after 2040? I don't see an explanation in the text on this.

Line 215 => It would be good to add some insights on implications for the power sector, and possibly also for CDR/BECCS/DAC and non-CO2 emissions (i.e., non-energy sector).

Lines 279-282 => This note on biofuels is a good example of inter-sectoral dynamics that could be highlighted more in this paper.

Lines 319-322 => Yes, this is very true. Please focus more on inter-sector dynamics.

Line 352 => So, apparently ammonia is not one of the four e-fuels modeled. What about methanol and other energy carriers. Please explain in the main text or methods section.

Line 424 => Why not compare to MAGICC, either in addition to FAIR or simply MAGICC alone? Does comparing to one or the other allow the GCAM scenarios to be more or less stringent, in terms of full-century carbon budgets?

Reviewer #2:

Remarks to the Author:

This paper provides the transport sector's global transition pathways to zero CO₂ emissions by 2050 using the GCAM global integrated assessment model. It indicates that electrification, biofuel and hydrogen penetration in all transport modes are key options to reach sectoral zero-emissions by 2050, as well as transport demand reduction. It concludes that deeper decarbonization in transport sector contributes to reducing mitigation challenges in other sectors, including industry and buildings sectors. Since the transport sector is recognized as one of the difficult to decarbonize sectors, the global zero-emissions pathways by 2050 in this study itself provides useful information to this research field in some extent. Nevertheless, I found few novel findings associated with energy and environmental implications, because they are already mentioned in many existing literatures. Also, discussion on feasibility of zero-emissions pathways in transport sector is lacking, it adds few scientific knowledge in its current form. Please see following comments for more detail.

- novelty

First, each energy option in zero -emissions scenario, such as electrification, biofuel and hydrogen (including e-fuel), is already well-known as they are quantitatively analyzed by the existing studies. The zero-emissions scenario assessed in this study simply enhances these options to reach sectoral zero emissions by 2050, without discussing on feasibilities and challenges associated with the additional mitigation options sufficiently. It is needed to enhance the novelty and the robustness of the zero-emissions scenarios.

Also, cost implication is lacking. Even if zero emissions in the transport sector are technically achievable, what is a challenge and how can it be compared with the mitigation options in other sectors? Though few cost information is provided in the discussion section, more detailed analysis and interpretations are needed. Also, additional infrastructure would be needed to decarbonize transport sector (e.g. EV charging facilities and hydrogen fueling station), but they are not mentioned in this paper. These mitigation challenges should be assessed and discussed further.

This paper concluded that further emission reductions in the transport sector result in moderate energy system changes in other sectors such as industry and transport, but such implication is not surprising in case the total emissions constraints are same across the scenario. As in the current form such analysis looks meaningless, thus I suggest reorganizing this section or dropping them.

- Scenario design

I found the concept of scenario design of three 1.5C scenarios not straightforward. Though I can understand the 1.5C high scenario is meaningful as such scenario was not included in the IPCC-AR6, the 1.5C low and medium transport scenarios are similar to some of scenarios included in the IPCC-AR6. In this regard, the current scenario design, assessing and comparing three 1.5C scenarios in parallel, looks inadequate to emphasize the novelty of this paper, thus I suggest reconsideration.

Also, the design of sensitivity scenarios which are provided in page 13-14 is insufficient. First, the purpose of sensitivity analysis is unclear. If one of the purposes is to assess the robustness of zero-emissions scenarios, additional factors that affect the transport energy system should be examined. For example, it can include pessimistic assumptions on cost reduction of key technologies (e.g. batteries for EV, hydrogen, renewables, etc.), resource scarcity and associated cost increase of battery, biofuel usage limitation associated with sustainability concerns, lock-in of long lifetime existing technology such as ship and airplanes. In addition, in terms of socio-economic uncertainty, I am not sure that SSP2 is adequate scenario for sensitivity analysis. At least should SSP5 be included?

- analysis

In the 1.5C-high scenario, oil-high-efficiency ship is increased rapidly around 2030 but phase out fast by 2040. Given its relatively long lifetime of ship, such behavior looks strange to me. Also, how the

GCAM model consider technology lock in? Please clarify. If it is not considered sufficiently, I consider this model not capable to assess the scenario with such drastic system changes.

Though electricity-based shipping and aviation are still a premature technology, they are increased after 2050 even in the reference case for short distance transport. What kind of technology does the model assume, and which source did you refer to set the parameter assumptions? I guess the information is provided in the external GCAM model description, but such important assumptions need to be provided in the manuscript as well.

Also, feasibility of some technology options needs to be discussed and justified further. Upscales of electric ship and plane in 1.5C-high scenario looks too optimistic.

In figure 3, total transport services in the 1.5C-high is slightly larger than 1.5C-low and medium across the mode. Please elaborate why such unintuitive results are observed.

I guess lower emissions in the 1.5C-high scenario result in economy-wide carbon price decreases, thus it affects to increase transport demand due to the price elasticity. If it is correct, such modeling framework does not make sense.

Though environmental implications are mentioned in the title, very limited information on environmental aspects is provided in the main text, while the paper mostly focuses on energy system implications. Only air pollutant emissions are provided in the supplementary table, but it looks insufficient. I suggest adding more analysis and interpretation of environmental aspects, or dropping "environmental implications" from the title of this paper.

I guess the GCAM model submitted a number of scenarios for the AR6. Given this situation, for the comparison with the AR6 scenario provided in Figure 1c, the AR6 scenarios submitted by GCAM and by other models should be distinguished for validation.

According to the table 1, modal shift to public transport is implemented for all 1.5C scenarios, but passenger rail and bus usage is decreasing in a longer term in Figure 2, while vehicle transport continues to increase. Please justify.

What does the "oil high efficiency" mean in some figures? If it refers to technological efficiency improvements of internal combustion engines, "biofuel high efficiency" or "e-fuel high efficiency" can be an option as well, but not appeared here.

- Specific comments.

Figure 4d

Hydrogen is used even in the reference scenario, does it make sense?

Carbon price effects are sometimes mentioned, but information on carbon price is not appeared in the figure. Please provide.

Line 311

Cost results are suddenly appeared here but I could not find corresponding results in the paper. Please provide them in the results section.

Figure 1

Please distinguish this study's three scenarios with different color or line type.

Figure 1

In terms of comparison with the AR6, only emissions are included, but comparison of energy consumptions would be useful. Please add in the main text or supplementary materials.

What does the passenger truck mean? Is it different with bus? Please clarify.

Figure S5

Does the cost here include the effect of carbon price? Please specify.

Reviewer #3:

Remarks to the Author:

I have uploaded my review as a Word document. Please reach out if you need me to provide some other way.

Nature communications Review

Review by Francis Vanek

Major comments: _____

This paper sets out to contribute to the literature by modeling the full range of global transportation modes and their emissions under different energy and emissions scenarios. In this reviewer's opinion, the objective is met and the paper should be published with some modest revisions, as detailed below. The pathways outlined, especially in the transportation high-ambition scenario, are very challenging indeed, but they are appropriate given the severity of the climate protection crisis.

I would like to see the revised manuscript once the authors have considered the comments below as well as those of other reviewers.

I have two major large comments. The first is in regard to the correlation developed in this paper between hydrogen as a fuel for shipping and biofuels as a fuel for aviation. I am particularly focused on the aviation side. I looked again at information from the two largest global manufacturers of commercial passenger aircraft (Airbus and Boeing) and it is clear to me that they are expending a considerable effort on the development of hydrogen as an alternative aviation fuel to the current hydrocarbon jet fuel. From what I can gather, Airbus and Boeing are developing this system not just for short-haul aviation but eventually for the entire market. This system is not a "hydrogen battery" as suggested in the manuscript but rather a fuel that functions in a similar way to conventional jet fuel, where the combustion of hydrogen drives a jet engine that is similar to current jets. This technology is a long way off in the future, but so is the development of a robust Fischer-Tropsch biofuel supply chain that can function at scale to meet most or all aviation needs. Since they are both futuristic, it would be prudent to allow that either one might eventually emerge as the aviation solution, rather than choosing the winner *a priori*. I don't think you need to revise all your simulations to include both biofuel and hydrogen options. Rather, the discussion could discuss the possibility (as a caveat) that hydrogen might eventually outvie biofuels to become the dominant aviation fuels. In any case, the various 2010-2100 figures throughout the paper would not change much if hydrogen were the aviation fuel rather than biofuels, except that one fuel replaces the other.

The second comment is to suggest that you might delve more deeply in the discussion at the end of the paper into the steps needed to "achieve deep decarbonization by developing low-carbon technologies" as you say on Page 2. It is implied by the paper, I think, that it is a matter of sufficient financial investment as to whether the low, medium, or high scenarios unfold. However, you might dig more deeply into all the elements that are required for such a massive scale-up or supply-chain problem. From our own writing, I can offer the following passage naming elements that are required for a successful scaleup (Vanek et al, 2014, p. 627):

1. Raw materials: Especially for building large quantities of fixed infrastructure and large numbers of vehicles, sources are required for various metals, concrete, glass, etc.
2. Manufacturing facilities: The total amount of infrastructure can expand year on year only as fast as the maximum output from manufacturing facilities will allow.
3. Locations for deployment: New locations must be found for the new infrastructure, which have met local approval and contribute to the solution in a meaningful way.
4. Financial capital: The financial system must be mobilized to provide sufficient funding to invest in the necessary infrastructure.
5. Human resources: At all stages of the supply chain, there must be sufficient numbers of people with the right skills in the work force to carry out each activity: financial management, resource extraction, manufacturing, freight shipment, installation, and maintenance.

The supply-chain management problem implies that shortfalls on any one of these points can greatly slow the evolution of the infrastructure: if any piece is missing, the infrastructure will not be able to expand to its full necessary capacity. Fortunately, in a properly functioning market economy, if there is need for new infrastructure, the market will respond by attempting to fill the need. For any of the above five areas, investment will follow demand. Take human resources as an example: if manufacturers and other employers are having difficulty finding enough skilled employees to make the various components of the new infrastructure, both government and educational institutions will respond to this gap. Governments can publicize the need for the skills and encourage entry into the workforce, and educational institutions can launch training programs and new degrees to meet the perceived need. Nevertheless, the rate at which these projects can be carried out may become the limiting factor in how quickly the infrastructure can be transformed. Similar arguments could be articulated for points 1 to 4 in the list.

A full discussion of this perspective on the supply chain implications of the transportation pathway scenarios is outside the scope of your paper; indeed, it would be worthy of a separate paper on its own. A passage in the discussion section would be sufficient to identify and briefly discuss the “scale-up” (aka “supply-chain”) problem.

Line-by-line comments

P1

L22 – Per above comment, you could drop “respectively”, i.e., “biofuels and hydrogen are particularly important for aviation and shipping.”

L23 – “reduce the pressure on other sectors to decarbonize:” I would delete this, all sectors (industrial and buildings as well as transportation) will be under pressure to decarbonize regardless.

P2

L37-38 An additional point worth mentioning here is the critical life-safety aspect of the fuel supply to maintain flight at cruising altitude. A ship on the ocean where the power train malfunctions can limp along, but an aircraft that encounters problems in the fuel supply puts lives at risk.

P3

L88-90 One could question the assertion that a GDP pathway can be chosen over the 2020-2100 time period independent of the rate of decarbonization. An inadequate decarbonization pathway could lead to radical destabilization of society (“the end of society as we know it,” to quote Al Gore), which in turn reduces GDP. A point perhaps for the discussion at the end of the paper.

P4

L100 Discussion of FT synthesis: This process is not yet a mature off-the-shelf option at the scale needed, so it may be the case that in the end hydrogen wins out over FT synthesis for shipping and commercial aviation, per my main discussion point.

Table 1: shipping and aviation column, could be expanded to include biofuels/e-fuels/**hydrogen**

P5

L112 I would drop “well below” and simply state “warming below 1.5°C.” I find the nuance of saying that warming could end up well below 1.5°C to be over-optimistic. The combination of rapid technological change and where needed carbon removal by 2100 require daunting amounts of ambition from the community of nations, and likely if we are able to come on under this threshold in 2100 it will not be by a large margin.

P6

L149 Again, the caveat is that the unfolding of technology might play out differently, aviation might also rely on hydrogen for a large fraction of the energy supply.

L153 I don’t think “hydrogen-powered batteries” is the correct term, or if it is, please provide a source. See my major comment about hydrogen in aviation.

P8

Fig 2 I don’t think “Coal” should appear in this figure as an actual option with its own color in the bar chart (although you yourselves say in the comment that it is so small that you cannot see it.). If this is coal used to fire steam locomotives, then this activity is anachronistic and its amount is surely so small that it is not worth including in the figure. Instead, delete it from the figure and in a footnote or in the caption explain that there was a very small amount of freight moved using coal, and provide the number of ton-km. I can’t think of what else this would be other than a steam locomotive burning coal, but if it is something else, please explain.

Also in Fig 2: Here and throughout, I think it is better to use the y-axis caption “trillion pass-km or ton-km.” It took me a moment to realize that “pass” as written means pass-km.

P9

L186 I typically use the word “activity” rather than “service.” Therefore, wherever you first introduce the word service, for clarity, could you add something like “service (i.e., the volume of passenger-km or ton-km) or else “service (also called “activity,” i.e., the volume of passenger-km or ton-km)?

P10

L218 Interesting point about reduced transportation ambition forcing other sectors to limit emissions

P13

L277-278 Line reads that passenger increases while freight decreases, but on looking at Fig 5c passenger appears to peak around 2070 and then fall while freight rises continually, can you elaborate?

P16

L319 “exploration of inter- and intra-sectoral dynamics is important etc.” – this is a key point and is well illustrated in the paper

L332 “Though electricity is the most important etc...” – because the increase in low-carbon electricity (e.g., from renewables) required is so large, this might be a good place to work in some discussion of what is needed to scale up ambition beyond a larger global financial investment. See my main comment above.

L335 You might mention that currently a large amount of industrial hydrogen is produced but almost entirely using steam reforming of methane (“grey hydrogen”), which emits CO₂. It is in my view legitimate to use some amount of grey hydrogen in the short run while the technologies are being worked out, even as infrastructure is being developed to deliver green or blue hydrogen.

L337 “hydrogen transmission and distribution networks:” It is in my view not a certainty that a hydrogen transmission network is a requirement. This infrastructure might instead use the electricity grid to move low-carbon electricity to the point of refueling hydrogen-powered vehicles (including ships and aircraft), so as to avoid the challenges of building out a hydrogen grid.

Reference:

Vanek, F, L Angenent, J Banks, R Daziano, and M Turnquist. (2014) *Sustainable Transportation Systems Engineering*. McGraw-Hill, New York.

** Reviewer comments in bold italics*

* Author responses in regular fonts

Reviewer #1 (Remarks to the Author):

This paper employs the global integrated assessment model GCAM to evaluate a variety of 1.5C-consistent scenarios for the transportation sector, considering varied timelines for the phase-out of fossil fuels and the implementation of advanced alternative technologies.

On the one hand, the model is top-of-class; the analysis is solid; and the results are interesting.

On the other hand, the methodological development here is minimal; the experimental set-up is not particularly creative; and the insights are far from surprising.

The authors state the following on p. 2:

“While there are a variety of industry scenarios and analyses for aviation and shipping decarbonization in isolation, these do not employ an integrated framework that considers the connections between the transportation sector and other sectors and environmental systems.”

This the real contribution that a paper of this type can make to the literature, failing a presentation of novel methodological developments. I would highly suggest that the author focus much more of their paper on systems interactions (both results and insights). The section starting on p. 10 get into this, but does not go far enough.

We thank the reviewer for their suggestion as to how to strengthen our paper and highlight our contribution to the existing literature. We agree that the ability to elucidate the interactions between the transportation sector and other energy sectors and environmental systems is a key strength of our modeling framework and approach, and have revised our paper to highlight these dynamics further. Specifically, we have split the previous Figure 4 (focused on systems interactions) into two figures (now Figures 4 and 5), incorporating the already existing analyses of emissions by sector and hydrogen and biofuels/e-fuels consumption by sector across the scenarios, as well as new panels showing carbon sequestration by sector (Figure 4d), electricity generation by technology (Figure 5c), and hydrogen generation by technology (Figure 5d). We expanded our section on “Impacts beyond the transportation sector” with discussion of these new figures, emphasizing the interplay between decarbonization ambition in the transportation sector and upstream consequences for fuel supply and downstream consequences for other sectors and emissions (lines 245-291, particularly new content on lines 255-262, 268-270, and 276-291). The section in the Supplementary Information, “Additional discussion of sensitivity scenarios,” also incorporates some consideration of these systems interactions; we have expanded this section with new sensitivity scenarios, including one in which economy-wide bioenergy consumption is more tightly constrained. We also added to the Discussion section on these topics, specifically lines 387-391, and highlight other portions of the Discussion (e.g., lines 408-410, 435-449) that also touch on systems interactions.

The quality of written English in the paper is strong, and the presentation and organization are very good.

Some more specific comments below...

Line 43 => The lit review in this and other sections is really excellent.

We thank the reviewer for their comment.

Line 94 => More information is needed to describe what exactly is an e-fuel in the GCAM context (with this modified version 6.0)? The Methods section contains a few words, but not enough.

We thank the reviewer for this comment and in response have expanded upon our description of e-fuels in the Methods section (lines 483-511) as follows:

“We added the capability to model synthetic hydrocarbon fuels using carbon captured from the atmosphere via direct air capture (DAC) and hydrogen as prospective drop-in replacements for today’s petroleum-derived liquid fuels. We parameterized four technologies for synthetic fuels production, which differ in how both the required electricity and hydrogen are produced. This is intended to represent varying degrees of stringency regarding the use of fossil fuels to generate the electricity and/or produce the hydrogen and allow each of these technologies to compete on a cost basis. The first technology uses grid electricity to capture CO₂ from the atmosphere and purchases an industrial hydrogen commodity which may either be produced on-site or delivered via pipeline or liquid truck from a centralized hydrogen production facility. The second technology uses grid electricity to capture CO₂ from the atmosphere and electrolyze hydrogen on-site. The last two technologies use electricity generated by dedicated wind turbines or solar panels, respectively, to run both the DAC and hydrogen electrolysis processes and are intended to represent the most restrictive definition for zero-carbon hydrogen and electricity sourcing. The levelized capital and fixed operating costs for the hydrogen electrolyzers and DAC equipment for these technologies are adjusted using regionally explicit capacity factors for wind turbines and solar panels and are harmonized with GCAM’s electricity sector^{74,75}. Parametric assumptions for hydrogen production and distribution are provided in the GCAM 6.0 release⁷³. The DAC cost and energy performance parameters assume a high-temperature, fully-electric liquid solvent-based process most similar to the one being developed by Carbon Engineering, as this is, to our knowledge, the most detailed publicly-available cost and performance data for a commercial DAC to liquid fuels process⁷⁶. The derivation of GCAM input assumptions for this process is documented in Fuhrman et al. (2021)⁷⁷. The energy and non-fuel cost coefficients for DAC are multiplied by 19.6 kg C per GJ of fuel, consistent with GCAM’s existing refined liquids commodity, and 1.19 GJ of hydrogen is assumed to be required per GJ of liquid fuels produced⁷⁸. Efforts to evaluate alternative sources of captured CO₂, including waste CO₂ and low-temperature DAC processes in GCAM are the subject of separate studies. We do not consider ammonia as a potential zero carbon fuel due to the large uncertainty regarding fugitive emissions that could further disrupt planetary boundaries for reactive nitrogen and fully negate any climate benefit achieved by avoiding CO₂ emissions⁶². Efforts to model DAC-to-methanol as

a means of decarbonizing the petrochemical sector are also left as an area for future work, as this technology is not considered in this study.”

Line 99 => There are other biofuel conversion pathways that do not lead to competition with food crops. Why only allow the FT-cellulosic pathway?

We appreciate the reviewer’s point about the existence of multiple biofuel production pathways that make use of non-food crops. While GCAM does incorporate some of these other pathways, including ethanol production from cellulosic sources, we choose to only allow the FT-cellulosic pathway in our alternative fuel mandates for aviation and shipping because existing research suggests that, of the options modeled by GCAM, this pathway generates fuel with the highest potential for use in aviation and shipping (see, e.g., Balcombe et al., 2019; Müller-Casseres et al., 2022). For example, Balcombe et al. (2019) note that FT-diesel could be used in existing marine diesel engines with only small alterations to the engines and could employ existing storage infrastructure, while use of bio-ethanols would necessitate larger-scale changes to both engines and fueling systems. We note the considerations around the suitability of fuel for use in aviation and shipping in lines 102-103.

Line 166 => In Figure 2, why are passenger rail and bus declining so quickly after 2040? I don't see an explanation in the text on this.

We thank the reviewer for their clarifying question. There are multiple factors that affect passenger rail and bus usage in the 1.5°C scenarios. Firstly, underlying all scenarios, the increasing global per capita GDP (Figure S2) facilitates a shift towards faster modes that have lower wait times; this is visible in the reference scenario, where usage of passenger rail and buses declines rapidly with time (see Figure S7). This declining service with time means that when the demand reduction assumptions are implemented in the 1.5°C scenarios, the resulting decrease in usage looks even more stark for passenger rail and buses (in comparison with passenger cars and trucks, for example, which see increasing usage until late century in the reference scenario). The elevated public transit preference that we implement in the 1.5°C scenarios acts to oppose the income-driven shift away from passenger rail and buses, but does not fully negate it. As a result of the public transit preference, the share of vehicle-kilometers provided by passenger rail and buses is higher in the 1.5°C scenarios than in the reference scenario (see new Figure S6). We discuss this now on lines 194-198 in the manuscript, with the accompanying supplemental Figure S6.

Line 215 => It would be good to add some insights on implications for the power sector, and possibly also for CDR/BECCS/DAC and non-CO2 emissions (i.e., non-energy sector).

We thank the reviewer for their suggestion and agree that these are useful insights to highlight. As mentioned in our response above, we have added new figures discussing implications for electricity and hydrogen production (Figures 5c and 5d) as well as a figure showing carbon sequestration by sector (Figure 4d) across the scenarios. We discuss these new figures on lines 255-262 (Figure 4d) and 280-291 (Figures 5c and 5d).

Lines 279-282 => This note on biofuels is a good example of inter-sectoral dynamics that could be highlighted more in this paper.

We thank the reviewer for their suggestion and point them to our detailed response to their general comment above. Our new figures 4d, 5c, and 5d address this, and are discussed in lines 245-291, particularly new content on lines 255-261, 268-270, and 276-291.

Lines 319-322 => Yes, this is very true. Please focus more on inter-sector dynamics.

We thank the reviewer for their suggestion and point them to our detailed response to their general comment above. Specifically, in the Discussion section, we added lines 387-391, and highlight other portions of the Discussion (e.g., lines 408-410, 435-449) that also touch on systems interactions.

Line 352 => So, apparently ammonia is not one of the four e-fuels modeled. What about methanol and other energy carriers. Please explain in the main text or methods section.

The reviewer is correct. For this study we do not model ammonia as a zero-carbon transportation fuel as preliminary analysis by our group (Wolfram et al., 2022) revealed the potential for large-scale disruptions to the global nitrogen cycle and potential further exacerbation of global warming if ammonia were to be used for marine shipping. We focused our e-fuels modeling on prospective drop-in replacements for existing liquid hydrocarbon fuels, although the use of DAC-to-methanol as a means for decarbonizing the petrochemical sector is the subject of a separate, forthcoming study by our group. We have clarified these points in the Methods section as described in response to the comment above.

Line 424 => Why not compare to MAGICC, either in addition to FAIR or simply MAGICC alone? Does comparing to one or the other allow the GCAM scenarios to be more or less stringent, in terms of full-century carbon budgets?

We thank the reviewer for their questions. In Figure S3, we compare the scenarios in our study to scenarios in the AR6 database in terms of their total CO₂ and greenhouse gas emissions and global mean temperature increase. For the temperature comparison, we use the output for global mean temperature increase from Hector, the reduced form climate model linked to GCAM, for our scenarios, and compare this to the 50th percentile FaIRv1.6.2 surface temperature variable for the AR6 scenarios (as described on original lines 424-425, now lines 536-538). We chose this variable for the AR6 scenarios because we also employ the “FaIRv1.6.2 category” for selecting scenarios to include in our comparison (i.e., to find scenarios in the database that are consistent with limiting end-of-century warming to 1.5°C), and wanted to maintain consistency across our variable selections and figures. As for why we use the “FaIRv1.6.2 category” rather than a comparable one for MAGICC, we found that more scenarios in the database included categorizations for the FaIR category. We show a comparison of how the temperature portion of Figure S3 (panel c) would look if we used the 50th percentile MAGICCv7.5.3 surface temperature variable instead of the 50th percentile FaIRv1.6.2 surface temperature variable in the figure below. If anything, it seems that the comparison with FaIR makes the GCAM scenarios look slightly warmer, particularly in earlier periods, but the differences are not notable for the 1.5°C-consistent scenarios.

a Global mean temperature increase compared to FaIR outputs from AR6

b Global mean temperature increase compared to MAGICC outputs from AR6

- AR6 reference: GCAM
- AR6 reference: other models
- AR6 1.5C: GCAM
- AR6 1.5C: other models
- Reference
- 1.5C low transport tech
- 1.5C medium transport tech
- 1.5C high transport tech

Reviewer #2 (Remarks to the Author):

This paper provides the transport sector's global transition pathways to zero CO2 emissions by 2050 using the GCAM global integrated assessment model. It indicates that electrification, biofuel and hydrogen penetration in all transport modes are key options to reach sectoral zero-emissions by 2050, as well as transport demand reduction. It concludes that deeper decarbonization in transport sector contributes to reducing mitigation challenges in other sectors, including industry and buildings sectors. Since the transport sector is recognized as one of the difficult to decarbonize sectors, the global zero-emissions pathways by 2050 in this study itself provides useful information to this research field in some extent. Nevertheless, I found few novel findings associated with energy and environmental implications, because they are already mentioned in many existing literatures. Also, discussion on feasibility of zero-emissions pathways in transport sector is lacking, it adds few scientific knowledge in its current form. Please see following comments for more detail.

- novelty

First, each energy option in zero -emissions scenario, such as electrification, biofuel and hydrogen (including e-fuel), is already well-known as they are quantitatively analyzed by the existing studies. The zero-emissions scenario assessed in this study simply enhances these options to reach sectoral zero emissions by 2050, without discussing on feasibilities and challenges associated with the additional mitigation options sufficiently. It is needed to enhance the novelty and the robustness of the zero-emissions scenarios.

We thank the reviewer for their comment. Our paper has dual goals: both novelty and synthesis. We still seek to provide novel results with our work, specifically with the 1.5°C high transport technology scenario, which incorporates drastic emissions reductions from the full transport sector beyond levels achieved in other studies and in existing scenarios from the IPCC AR6 report. However, an equally important ambition of our study is to offer a synthesis of a range of possible outcomes for the transportation sector in a 1.5°C-consistent future. While the reviewer is correct that we do not offer any “new” technology options in our work, we incorporate each of these technology options for all the transportation modes into a single integrated framework. This allows us to explore the collective impact of these technologies and the economy-wide implications of different pathways for transportation decarbonization. As we outline in the introduction section of our paper, the existing literature is lacking a global, integrated assessment of deep decarbonization pathways for the full transportation sector that evaluates the relative contributions of different technology options for each of the transport modes; we aim to fill this gap with our work.

With regards to the comment about feasibility issues and challenges facing achieving ambitious levels of transportation decarbonization, we have revised our manuscript to further emphasize these topics. We provide more details in response to the questions below, but lines 419-454 in the Discussion section now discuss these issues, and we have incorporated metrics relating to the scale of potential costs associated with the transition on lines 180-189 in the Results section.

Also, cost implication is lacking. Even if zero emissions in the transport sector are technically achievable, what is a challenge and how can it be compared with the mitigation options in other sectors? Though few cost information is provided in the discussion section, more detailed analysis and interpretations are needed. Also, additional infrastructure would be needed to decarbonize transport sector (e.g. EV charging facilities and hydrogen fueling station), but they are not mentioned in this paper. These mitigation challenges should be assessed and discussed further.

We thank the reviewer for their comments and have revised our paper to address both of these considerations. We include additional cost implications in the Results section on lines 180-189 and Table S4, where we discuss the break-even carbon prices that would be required to make electric and hydrogen-based shipping and aviation technologies economically competitive. We have also restructured our Discussion section to place more prominence on our consideration of the challenges facing realization of the rapid technological transitions in the transport sector that are required to facilitate deep decarbonization. We now discuss these issues in detail on lines 419-449, particularly lines 426-449, and highlight specifically the infrastructure needed for electric and hydrogen-powered vehicles on lines 430-435: “High levels of transport electrification would necessitate the widespread deployment of charging infrastructure, for vehicles ranging from passenger cars to large cargo ships, and produce a large demand for critical minerals for battery development^{10,31}. Similarly, ubiquitous transportation hydrogen use would require the large-scale development of vehicle technologies, as well as hydrogen transmission and distribution networks and/or on-site electrolysis systems at refueling stations^{10,49}.” We also point to the cost, infrastructure, and supply chain issues as areas for future research on lines 450-454.

This paper concluded that further emission reductions in the transport sector result in moderate energy system changes in other sectors such as industry and transport, but such implication is not surprising in case the total emissions constraints are same across the scenario. As in the current form such analysis looks meaningless, thus I suggest reorganizing this section or dropping them.

We thank the reviewer for their comment. While we agree that it is expected that other sectors would respond to differing levels of ambition in transport, since overall emissions constraints are held constant, we see value in identifying which specific sectors are most affected and in highlighting the details of the systems interactions that emerge. Thus, the novelty is not in pointing out that there are trade-offs between different transportation decarbonization scenarios, but rather in highlighting what specifically the most notable of these trade-offs are and how they manifest, which we aim to do in lines 245-291. Furthermore, as the other two reviewers identified our analysis of these systems dynamics as one of the most valuable portions of our paper, we have chosen to retain this analysis. However, based on this comment as well as the suggestion of another reviewer, we have deepened our consideration of the interactions between the transportation sector and other sectors and systems by including new figures on carbon sequestration (Figure 4d) and electricity and hydrogen production

(Figures 5c and 5d) in our scenarios. These new figures and accompanying comments (particularly new content on 252-259, 265-267, and 273-288) allow us to further evaluate both upstream and downstream implications of different levels of ambition in the transportation sector.

- Scenario design

I found the concept of scenario design of three 1.5C scenarios not straightforward. Though I can understand the 1.5C high scenario is meaningful as such scenario was not included in the IPCC-AR6, the 1.5C low and medium transport scenarios are similar to some of scenarios included in the IPCC-AR6. In this regard, the current scenario design, assessing and comparing three 1.5C scenarios in parallel, looks inadequate to emphasize the novelty of this paper, thus I suggest reconsideration.

While we appreciate the reviewer's point about how the 1.5°C low and medium transport technology scenarios are within the range of the scenarios included in the IPCC AR6 report, we still find utility in including them in our analysis. With our paper, we are aiming not just to include novel results—i.e., the 1.5°C high transport technology scenario, which represents a level of ambition in the transportation sector not currently present in the IPCC AR6 report—but also to provide a synthesis of potential futures for the transportation sector in a 1.5°C-consistent world and their broader scale implications. The low and medium transport technology scenarios represent the more conventional view of the sector as “difficult to decarbonize,” expected to maintain residual emissions even once economy-wide net zero is achieved. Incorporating these scenarios alongside our novel, high ambition scenario in our integrated framework allows us to evaluate the implications of different levels of ambition in the transportation sector for other elements of the energy system and the net zero transition (see lines 71-75 in the introduction section justifying this framework). Thus, while we highlight the unique contribution of our high ambition scenario in the first part of the paper (e.g., Figures 1 and 2), we also spend a significant portion of the paper discussing the differences between high, medium, and low ambition scenarios for the transportation sector, both in terms of technological transitions within the sector and in terms of their integrated impacts on energy demands and the power system, and required levels of decarbonization ambition in other sectors (e.g., Figures 3-5).

Also, the design of sensitivity scenarios which are provided in page 13-14 is insufficient. First, the purpose of sensitivity analysis is unclear. If one of the purposes is to assess the robustness of zero-emissions scenarios, additional factors that affect the transport energy system should be examined. For example, it can include pessimistic assumptions on cost reduction of key technologies (e.g. batteries for EV, hydrogen, renewables, etc.), resource scarcity and associated cost increase of battery, biofuel usage limitation associated with sustainability concerns, lock-in of long lifetime existing technology such as ship and airplanes. In addition, in terms of socio-economic uncertainty, I am not sure that SSP2 is adequate scenario for sensitivity analysis. At least should SSP5 be included?

We thank the reviewer for their suggestion about expanding our sensitivity analysis. We agree that further analysis of factors that impact the transportation system would be beneficial;

while we cannot address all sensitivities in this paper due to space limitations, we have incorporated additional socioeconomic sensitivity scenarios (now including all SSPs 2-5) as well as a scenario in which biofuel use is even more tightly constrained than in our standard 1.5°C scenarios. These new scenarios appear in the revised Figure 6 and are discussed on lines 324-333, 343-360, in the Methods on lines 568-582, and in the now expanded Supplementary Information section “Additional discussion of sensitivity scenarios.” We have also further explained the justification for our sensitivity analysis in lines 318-323, highlighting that we aim to evaluate the robustness of our results, specifically by assessing the feasibility of implementing the high ambition technological transition in the transportation sector and meeting the same decarbonization targets under less optimistic assumptions for socioeconomic development pathways and societal willingness to cut back on transportation usage to facilitate meeting climate goals (as these assumptions underlie all of our scenarios).

- analysis

In the 1.5C-high scenario, oil-high-efficiency ship is increased rapidly around 2030 but phase out fast by 2040. Given its relatively long lifetime of ship, such behavior looks strange to me. Also, how the GCAM model consider technology lock in? Please clarify. If it is not considered sufficiently, I consider this model not capable to assess the scenario with such drastic system changes.

We thank the reviewer for their comment and have a few remarks to make in response. Firstly, the “oil high efficiency” category is intended to represent efficiency improvements to standard oil-based engines, rather than a separate technology itself. Thus, the rapid increase in oil high efficiency ships should not be seen as requiring a full technological transition, but rather fleet average efficiency improvements. However, as we mention in response to another question from the reviewer below, since the efficiency changes are not a focus of our study, we have now changed our figures to aggregate the standard and high efficiency technologies together when showing transportation service by technology (and added a comment on this in lines 476-480).

Secondly, while GCAM does not directly consider technology lock-in for ships, we take into account the typical lifetimes of marine vessels and premature retirement when designing our scenarios and setting the low carbon technology transition timelines. The rapid transitions we see in the high transport technology scenario demonstrate the difficulty of achieving the most aggressive fossil phase-out targets in the transport sector. Additionally, our results can provide insight into stranded asset risk for modes that are not vintaged, such as marine shipping. For domestic shipping, the shift to electric ships occurs rapidly; our results can therefore be taken as an indication that domestic shipping shows high potential for electrification but also faces a high risk of oil-based vessels becoming stranded assets, if the most ambitious targets for transportation decarbonization are to be achieved. For international shipping, the most immediate transitions are to biofuel-based service, while electric and hydrogen-based service take more time to scale up. As biofuels and other alternative liquid fuels are being designed largely as drop-in fuels that could function with retrofits to existing engine technologies, the transitions in international shipping should thus be more feasible with lower risk of assets being compromised.

We have added a comment on the speed of the transition for domestic shipping in particular on lines 176-179: “In domestic shipping, electricity is dominant, supplying over 80% of service from 2050 onwards in the high scenario; the speed with which this transition occurs implies both a high potential for electrification and an elevated risk of short-haul freight vessels becoming stranded assets.”

Though electricity-based shipping and aviation are still a premature technology, they are increased after 2050 even in the reference case for short distance transport. What kind of technology does the model assume, and which source did you refer to set the parameter assumptions? I guess the information is provided in the external GCAM model description, but such important assumptions need to be provided in the manuscript as well.

We thank the reviewer for their questions and point them to the section in the Supplementary Information titled “Shipping and aviation technology assumptions” (referenced on lines 470-473 in the main manuscript) in which we provide a detailed table describing the non-fuel costs and energy intensities assumed for shipping and aviation technologies. Based on the reviewer’s questions, we have elaborated on our description of these technologies as well as our explanation of the sources used to set the parameter assumptions:

“Electric and hydrogen-powered aviation and shipping costs are based on existing literature³⁻⁵, where available, or are estimated based on the cost ratio between conventional and advanced technologies for modes with similar drivetrains. Specifically, battery electric aircraft costs, including both capital costs and non-fuel operation and maintenance costs, are obtained from the all-electric aircraft technology outlined in Schäfer et al.⁴ This technology is assumed to have a fuel efficiency double that of standard jet engine planes. Hydrogen-powered aircraft are modeled after the technologies outlined in the Destination 2050 report and represent aircraft that combust hydrogen in a turbine^{3,5}. This technology is expected to increase capital costs by 31% and non-fuel operation and maintenance costs by 47% relative to conventional aircraft due to the costs associated with the hydrogen storage tank and the fuel distribution system, as well as the larger required aircraft size^{3,5}. Hydrogen-powered aircraft are also projected to have longer refueling times than conventional aircraft, decreasing flight cycles by 7%, as well as lowered seating capacity, further decreasing productivity by about 12%^{3,5}. For marine shipping vessels, elevated costs for battery electric and fuel cell electric technologies are calculated based on the cost increases associated with the corresponding technologies in freight rail, as ships and rail have similar drivetrain technologies. Specifically, the ratio of battery electric rail costs to conventional rail costs is applied to conventional ship costs to obtain battery electric ship costs (with a comparable calculation performed for fuel cell electric ship costs). For freight rail, advanced technology costs are calculated from the rail technology assessment by the California Air Resources Board^{1,6}.”

For full documentation of the assumptions employed to parametrize the advanced transportation technologies in GCAM, for all modes in addition to shipping and aviation, we refer the readers to the GCAM documentation and relevant transportation-specific documentation files^{1,2}.

References:

1. Kyle, P., Fuhrman, J., Wolfram, P., O'Rourke, P. & Kholod, N. Core Model Proposal #359: Hydrogen and transportation technology update. (2022).
2. Mishra, G. S. et al. Transportation Module of Global Change Assessment Model (GCAM). (2013).
3. Royal Netherlands Aerospace Centre. Destination 2050: A Route To Net Zero European Aviation. <https://reports.nlr.nl/server/api/core/bitstreams/c9002b7e-224f-420c-b6da-ab6aecd48ea2/content> (2021).
4. Schäfer, A. W. et al. Technological, economic and environmental prospects of all-electric aircraft. *Nat Energy* 4, 160–166 (2019).
5. McKinsey & Company. Hydrogen-powered aviation: A fact-based study of hydrogen technology, economics, and climate impact by 2050. https://www.euractiv.com/wp-content/uploads/sites/2/2020/06/20200507_Hydrogen-Powered-Aviation-report_FINAL-web-ID-8706035.pdf (2020).
6. California Air Resources Board. Technology Assessment: Freight Locomotives. https://ww2.arb.ca.gov/sites/default/files/classic/msprog/tech/techreport/final_rail_tech_assessment_11282016.pdf (2016).

Also, feasibility of some technology options needs to be discussed and justified further. Upscales of electric ship and plane in 1.5C-high scenario looks too optimistic.

We agree that the scale-up of electric and hydrogen-based technologies in the 1.5°C high transport technology scenario is indeed ambitious. This is part of the reason why we have the high/medium/low scenario framework, to be able to explore different degrees of ambition; we are not predicting what we expect to happen with any of the scenarios, and in particular not with the high scenario, but rather are intending to demonstrate the technology transitions that would be necessary if the transport sector is to achieve gross zero emissions by 2050. We also note that the levels of mitigation in other sectors and/or carbon capture utilization that would be required to achieve economy-wide net zero emissions by 2050 without such ambitious transport decarbonization are themselves also optimistic and ambitious. With such long-term modeling, what is classified as too optimistic is ambiguous and somewhat subjective.

However, we appreciate the point that the rapid electrification of ships and planes in our high ambition scenario will require massive, systems-level change, and face substantial barriers; we have commented on these issues in our revised Discussion section (lines 426-449, in particular the note on the nascency of low carbon technologies for shipping and aviation in lines 427-430).

In figure 3, total transport services in the 1.5C-high is slightly larger than 1.5C-low and medium across the mode. Please elaborate why such unintuitive results are observed. I guess lower emissions in the 1.5C-high scenario result in economy-wide carbon price decreases, thus it affects to increase transport demand due to the price elasticity. If it is correct, such modeling framework does not make sense.

We thank the reviewer for their question. We discuss the differences in transportation service provision between the scenarios on lines 222-238 as well as in the Supplementary

Information section “Additional discussion of shifting patterns of transportation service across scenarios,” and have elaborated on our commentary on this in both sections. As we describe there, meeting emissions mitigation targets in the 1.5°C low transport technology scenario, in which fuel switching is difficult, requires elevated carbon prices and a reliance on demand destruction in the transport sector. In contrast, in the 1.5°C high transport technology scenario, the more aggressive deployment of low carbon technologies enables more fuel switching than in the low scenario and thus results in lower carbon prices and a lessened dependence on reductions in service to meet emissions mitigation targets.

Though environmental implications are mentioned in the title, very limited information on environmental aspects is provided in the main text, while the paper mostly focuses on energy system implications. Only air pollutant emissions are provided in the supplementary table, but it looks insufficient. I suggest adding more analysis and interpretation of environmental aspects, or dropping “environmental implications” from the title of this paper.

We appreciate the reviewer’s point and have changed the title of the paper to “A zero-emissions global transportation sector: Integrated assessment modeling of advanced technologies.”

I guess the GCAM model submitted a number of scenarios for the AR6. Given this situation, for the comparison with the AR6 scenario provided in Figure 1c, the AR6 scenarios submitted by GCAM and by other models should be distinguished for validation.

We thank the reviewer for their suggestion and agree that it is beneficial to distinguish which of the scenarios from the AR6 database came from GCAM. We have revised Figure 1c to now feature the GCAM-AR6 scenarios in a different color than the AR6 scenarios generated by other models. We incorporated similar differentiation in colors in Figure S3 as well.

According to the table 1, modal shift to public transport is implemented for all 1.5C scenarios, but passenger rail and bus usage is decreasing in a longer term in Figure 2, while vehicle transport continues to increase. Please justify.

We thank the reviewer for their clarifying question. There are multiple factors that affect passenger rail and bus usage in the 1.5°C scenarios. Firstly, underlying all scenarios, the increasing global per capita GDP (Figure S2) facilitates a shift towards faster modes that have lower wait times; this is visible in the reference scenario, where usage of passenger rail and buses declines rapidly with time (see Figure S7). This declining service with time means that when the demand reduction assumptions are implemented in the 1.5°C scenarios, the resulting decrease in usage looks even more stark for passenger rail and buses (in comparison with passenger cars and trucks, for example, which see increasing usage until late century in the reference scenario). The elevated public transit preference that we implement in the 1.5°C

scenarios acts to oppose the income-driven shift away from passenger rail and buses, but does not fully negate it. As a result of the public transit preference, the share of vehicle-kilometers provided by passenger rail and buses is higher in the 1.5°C scenarios than in the reference scenario (see new Figure S6). Note that the share of vehicle-kilometers is a better indicator to look at here than the share of passenger-kilometers, because we also implement elevated ridesharing in our 1.5°C scenarios, which increases the passenger-kilometers provided by passenger road vehicles without increasing their vehicle-kilometers. This is part of why total passenger-kilometers do not decline as rapidly as would be expected in our 1.5°C scenarios.

We discuss this now on lines 194-198 in the manuscript, with the accompanying supplemental Figure S6. We also describe the implementation of our assumptions for elevated public transit preference, increased ridesharing, and demand reduction in the Methods section (lines 548-555).

What does the “oil high efficiency” mean in some figures? If it refers to technological efficiency improvements of internal combustion engines, “biofuel high efficiency” or “e-fuel high efficiency” can be an option as well, but not appeared here.

We thank the reviewer for their clarifying question. The “oil high efficiency” technology option indeed refers to technological efficiency improvements of internal combustion engines. GCAM includes a representation of high efficiency technologies for refined liquids-based transportation technologies; as the reviewer correctly points out, this applies to biofuels and e-fuels as well as to oil. We had previously chosen to only highlight the high efficiency technology for oil, aggregating high efficiency biofuel and e-fuel technologies with their corresponding standard efficiency counterparts. However, we appreciate the reviewer’s point about how this creates confusion. Since the efficiency changes are not a focus of our study, we have now changed our figures to aggregate the standard and high efficiency technologies together for all of the refined liquids, including oil, when showing transportation service by technology. We also added a comment on this in lines 476-480.

- Specific comments.

Figure 4d

Hydrogen is used even in the reference scenario, does it make sense?

We thank the reviewer for their comment. While we recognize the uncertainty associated with hydrogen deployment, our results indicate that anticipated future reductions in cost make hydrogen applicable in niche use cases even in the reference scenario, such as for some industrial and transport applications. We also note that while the magnitude of hydrogen utilization in transport and industry is comparable in the reference scenarios and the decarbonization scenarios (Figure 5a), the decarbonization scenarios incorporate demand reduction assumptions and thus hydrogen contributes a much larger share of the total energy consumption by these sectors in the decarbonization scenarios. For example, by 2050, hydrogen constitutes 6-24% of the fuel used in the transport sector in the decarbonization scenarios, but just 4% in the reference scenario; by 2100, these differences are even starker,

with 16-27% of transport energy coming from hydrogen in the decarbonization scenarios, relative to only 7% in the reference scenario (see Figure S9).

Carbon price effects are sometimes mentioned, but information on carbon price is not appeared in the figure. Please provide.

We thank the reviewer for this suggestion and have added a new figure on carbon prices across the decarbonization scenarios (Figure S10) which we reference in the text on line 230.

Line 311

Cost results are suddenly appeared here but I could not find corresponding results in the paper. Please provide them in the results section.

We thank the reviewer for their suggestion and have moved the cost results previously referred to in the Discussion to the Results section (lines 122-125), with corresponding reference to Supplemental Information Table S1. We have also added additional cost results (lines 180-189 and Table S4) specific to the shipping and aviation low carbon technologies.

Figure 1

Please distinguish this study's three scenarios with different color or line type.

We thank the reviewer for their suggestion to improve the clarity of our figure. We have now changed Figure 1 to include different colors for each of the scenarios in this study.

Figure 1

In terms of comparison with the AR6, only emissions are included, but comparison of energy consumptions would be useful. Please add in the main text or supplementary materials.

We thank the reviewer for this suggestion and have added a figure to the Supplementary Information (Figure S21) showing a comparison of energy use by the transport sector in our scenarios and in the scenarios in the AR6 database, i.e., a comparable version of Figure 1 but for energy use.

What does the passenger truck mean? Is it different with bus? Please clarify.

Passenger trucks refer to large cars or trucks that are used for personal, passenger use, rather than hauling. They include light trucks such as pickup trucks as well as SUVs. We have added a clarifying note explaining the “passenger cars and trucks” category to the description of Table 1 (lines 108-109), indicating that the category includes all passenger light duty vehicles.

Figure S5

Does the cost here include the effect of carbon price? Please specify.

We thank the reviewer for their clarifying question. The costs shown in Figure S5 do incorporate carbon price effects. We have now indicated this in the figure caption.

Reviewer #3 (Remarks to the Author):

Major comments:

This paper sets out to contribute to the literature by modeling the full range of global transportation modes and their emissions under different energy and emissions scenarios. In this reviewer's opinion, the objective is met and the paper should be published with some modest revisions, as detailed below. The pathways outlined, especially in the transportation high-ambition scenario, are very challenging indeed, but they are appropriate given the severity of the climate protection crisis.

I would like to see the revised manuscript once the authors have considered the comments below as well as those of other reviewers.

I have two major large comments. The first is in regard to the correlation developed in this paper between hydrogen as a fuel for shipping and biofuels as a fuel for aviation. I am particularly focused on the aviation side. I looked again at information from the two largest global manufacturers of commercial passenger aircraft (Airbus and Boeing) and it is clear to me that they are expending a considerable effort on the development of hydrogen as an alternative aviation fuel to the current hydrocarbon jet fuel. From what I can gather, Airbus and Boeing are developing this system not just for short-haul aviation but eventually for the entire market. This system is not a "hydrogen battery" as suggested in the manuscript but rather a fuel that functions in a similar way to conventional jet fuel, where the combustion of hydrogen drives a jet engine that is similar to current jets. This technology is a long way off in the future, but so is the development of a robust Fischer-Tropsch biofuel supply chain that can function at scale to meet most or all aviation needs. Since they are both futuristic, it would be prudent to allow that either one might eventually emerge as the aviation solution, rather than choosing the winner a priori. I don't think you need to revise all your simulations to include both biofuel and hydrogen options. Rather, the discussion could discuss the possibility (as a caveat) that hydrogen might eventually outvie biofuels to become the dominant aviation fuels. In any case, the various 2010-2100 figures throughout the paper would not change much if hydrogen were the aviation fuel rather than biofuels, except that one fuel replaces the other.

We thank the reviewer for their thoughtful comment. Firstly, we would like to clarify that our representation of a hydrogen-based aviation technology does indeed model aircraft using hydrogen combustion turbines (and not hydrogen fuel cells). We apologize for the confusion and have revised the manuscript to ensure that our description of the hydrogen technology for aviation is clear (e.g., revised lines 167-168 and Table 1, and added to the captions of Figures 2 and 3 the statement: "For aviation, hydrogen technologies employ hydrogen combustion turbines; for all other modes, hydrogen fuel cell electric vehicles are modeled.").

Secondly, we appreciate the reviewer's point about the significant uncertainties associated with both FT biofuel and hydrogen technologies for aviation, and agree that

investments in these technologies in the coming years could greatly alter their trajectories and potential as competitive decarbonization options. We have added a caveat to the Discussion section (lines 400-406) highlighting this uncertainty: “While we observe that the decarbonization of long-haul aviation heavily depends on biofuel use, synthesis of biofuels via the FT pathway will still require advanced research, development, and commercialization to be used on the scale necessary to satisfy aviation biofuel demand^{48,57}. Given these challenges, as well as potential limitations on bioenergy use and the increasing investments in the development of hydrogen propulsion systems for aviation by some industry leaders⁵⁸, it is possible that hydrogen-based technologies could outpace biofuels to provide a larger share of long-haul aviation service than indicated by our results.”

The second comment is to suggest that you might delve more deeply in the discussion at the end of the paper into the steps needed to “achieve deep decarbonization by developing low-carbon technologies” as you say on Page 2. It is implied by the paper, I think, that it is a matter of sufficient financial investment as to whether the low, medium, or high scenarios unfold. However, you might dig more deeply into all the elements that are required for such a massive scale-up or supply-chain problem. From our own writing, I can offer the following passage naming elements that are required for a successful scaleup (Vanek et al, 2014, p. 627):

- 1. Raw materials: Especially for building large quantities of fixed infrastructure and large numbers of vehicles, sources are required for various metals, concrete, glass, etc.***
- 2. Manufacturing facilities: The total amount of infrastructure can expand year on year only as fast as the maximum output from manufacturing facilities will allow.***
- 3. Locations for deployment: New locations must be found for the new infrastructure, which have met local approval and contribute to the solution in a meaningful way.***
- 4. Financial capital: The financial system must be mobilized to provide sufficient funding to invest in the necessary infrastructure.***
- 5. Human resources: At all stages of the supply chain, there must be sufficient numbers of people with the right skills in the work force to carry out each activity: financial management, resource extraction, manufacturing, freight shipment, installation, and maintenance.***

The supply-chain management problem implies that shortfalls on any one of these points can greatly slow the evolution of the infrastructure: if any piece is missing, the infrastructure will not be able to expand to its full necessary capacity. Fortunately, in a properly functioning market economy, if there is need for new infrastructure, the market will respond by attempting to fill the need. For any of the above five areas, investment will follow demand. Take human resources as an example: if manufacturers and other employers are having difficulty finding enough skilled employees to make the various components of the new infrastructure, both government and educational institutions will respond to this gap. Governments can publicize the need for the skills and encourage entry into the workforce, and educational institutions can launch training programs and new degrees to meet the perceived need. Nevertheless, the rate at which these projects can be carried out may become the limiting factor in how quickly the infrastructure can be transformed. Similar arguments could be articulated for points 1 to 4 in the list.

A full discussion of this perspective on the supply chain implications of the transportation pathway scenarios is outside the scope of your paper; indeed, it would be worthy of a separate paper on its own. A passage in the discussion section would be sufficient to identify and briefly discuss the “scale-up” (aka “supply-chain”) problem.

We thank the reviewer for this thoughtful suggestion on expanding our Discussion section. Based on the comments of this reviewer and Reviewer 2, we have restructured our Discussion section to place more prominence on our consideration of the challenges facing realization of the rapid technological transitions in the transport sector that we identify in our work. We now discuss this in detail on lines 419-449, particularly lines 426-449, and explicitly identify the challenges raised at each stage of the supply chain (as highlighted by the reviewer) in lines 445-449: “For all low carbon technologies, ensuring that sufficient financial capital and human resources are available at all stages of the supply chain—including procurement of raw materials and resources, technology manufacturing and deployment, and infrastructure siting and development—will be critical to avoid any bottlenecks that could constrain or impede the massive scale-up required^{31,68}.” We also highlight these topics as areas to explore in future research in lines 451-454; indeed, members of our research group have plans for future papers evaluating some of these issues, particularly those related to raw materials, finance, and human resources.

Line-by-line comments

P1

L22 – Per above comment, you could drop “respectively”, i.e., “biofuels and hydrogen are particularly important for aviation and shipping.”

We thank the reviewer for this suggestion and have made this change (line 24).

L23 – “reduce the pressure on other sectors to decarbonize:” I would delete this, all sectors (industrial and buildings as well as transportation) will be under pressure to decarbonize regardless.

We thank the reviewer for this suggestion and have revised this line to read “While increased emissions mitigation in the transportation sector contributes substantially to achieving climate targets, the required rapid technological shifts have implications for resource demands and fuel availability” (lines 24-26).

P2

L37-38 An additional point worth mentioning here is the critical life-safety aspect of the fuel supply to maintain flight at cruising altitude. A ship on the ocean where the power train malfunctions can limp along, but an aircraft that encounters problems in the fuel supply puts lives at risk.

We appreciate the reviewer noting this consideration and have incorporated it into this section. Lines 36-38 (previous lines 37-38) now read: “Efforts to rapidly decarbonize these modes are further complicated by long vehicle lifespans and, for aviation in particular, the need for a dependable fuel supply that can ensure safety and sustain operations at cruising altitude^{2,10}.”

P3

L88-90 One could question the assertion that a GDP pathway can be chosen over the 2020-2100 time period independent of the rate of decarbonization. An inadequate decarbonization pathway could lead to radical destabilization of society (“the end of society as we know it,” to quote Al Gore), which in turn reduces GDP. A point perhaps for the discussion at the end of the paper.

We agree that there is likely to be a strong interconnection between the global GDP pathway and decarbonization trajectory, and appreciate the point that failure to meet climate goals could have repercussions for GDP (and society at large). However, due to space considerations, and as these issues are largely outside the scope of our paper, we did not incorporate this point into the Discussion section, instead reserving the space for an expanded evaluation of the scale-up and feasibility issues associated with the high ambition transportation decarbonization scenario and other integrated systems impacts of varied levels of ambition in the transport sector.

P4

L100 Discussion of FT synthesis: This process is not yet a mature off-the-shelf option at the scale needed, so it may be the case that in the end hydrogen wins out over FT synthesis for shipping and commercial aviation, per my main discussion point.

We thank the reviewer for noting this, and have added a caveat to the Discussion section (lines 400-406) highlighting this uncertainty: “While we observe that the decarbonization of long-haul aviation heavily depends on biofuel use, synthesis of biofuels via the FT pathway will still require advanced research, development, and commercialization to be used on the scale necessary to satisfy aviation biofuel demand^{48,57}. Given these challenges, as well as potential limitations on bioenergy use and the increasing investments in the development of hydrogen propulsion systems for aviation by some industry leaders⁵⁸, it is possible that hydrogen-based technologies could outpace biofuels to provide a larger share of long-haul aviation service than indicated by our results.”

Table 1: shipping and aviation column, could be expanded to include biofuels/e-fuels/hydrogen

We thank the reviewer for this suggestion and have adjusted our description of the scenarios in Table 1 to further highlight the role of biofuels/e-fuels, as well as electric and hydrogen-based technologies.

P5

L112 I would drop “well below” and simply state “warming below 1.5°C.” I find the nuance of saying that warming could end up well below 1.5°C to be over-optimistic. The combination of rapid technological change and where needed carbon removal by 2100 require daunting amounts of ambition from the community of nations, and likely if we are able to come on under this threshold in 2100 it will not be by a large margin.

We appreciate this point from the reviewer and have revised this statement to say “All decarbonization scenarios achieve deep economy-wide emissions reductions, bringing global mean warming below 1.5°C by the end of the century” (lines 115-116).

P6

L149 Again, the caveat is that the unfolding of technology might play out differently, aviation might also rely on hydrogen for a large fraction of the energy supply.

We thank the reviewer for this comment and again point them to the caveat on this that we have added to the Discussion section (lines 400-406). We have also revised the description of service provision for long-haul aviation in the Results section to highlight this caveat as well; lines 167-170 now read: “In long-haul aviation, the high costs of using electric batteries or hydrogen combustion systems on long haul flights—barring significant investments in and advancements of these technologies—prevent both technologies from taking off, leading to a dependence on alternative liquid fuels to meet decarbonization targets.”

L153 I don’t think “hydrogen-powered batteries” is the correct term, or if it is, please provide a source. See my major comment about hydrogen in aviation.

We thank the reviewer for pointing this out, as they are correct in noting that “hydrogen-powered batteries” was indeed not the appropriate phrasing. In GCAM, we model a hydrogen-based jet propulsion system for aviation, which differs from the hydrogen fuel cell vehicles modeled for the other transport modes. We have revised this line to now read “In long-haul aviation, the high costs of using electric batteries or hydrogen combustion systems on long haul flights...” (lines 167-168) and have clarified this in Table 1 and the captions of Figures 2 and 3.

P8

Fig 2 I don’t think “Coal” should appear in this figure as an actual option with its own color in the bar chart (although you yourselves say in the comment that it is so small that you cannot see it.). If this is coal used to fire steam locomotives, then this activity is anachronistic and its amount is surely so small that it is not worth including in the figure. Instead, delete it from the figure and in a footnote or in the caption explain that there was a very small amount of freight moved using coal, and provide the number of ton-km. I can’t think of what else this would be other than a steam locomotive burning coal, but if it is something else, please explain.

We thank the reviewer for their suggestion and agree that removing coal from the figure enhances clarity. We have removed coal as an option from the figure and legend and now include a statement in the caption of Figure 2 explaining the limited use of coal in transport: “Also note that there is a very small amount of coal-based freight rail service that is not shown in the figure as it phases out by 2025 and provides only a miniscule contribution to global freight rail transport in the preceding periods (0.009 trillion ton-km of service in 2005, 0.007 trillion ton-km in 2010, and 0.002 trillion ton-km in 2015 and 2020)” (lines 205-208). We have similarly removed coal as a legend option in Figure 6 and added a corresponding note there, and have done the same for relevant supplemental figures.

Also in Fig 2: Here and throughout, I think it is better to use the y-axis caption “trillion pass-km or ton-km.” It took me a moment to realize that “pass” as written means pass-km.

We thank the reviewer for their suggestion to increase the clarity of our axis labels and have changed all the relevant axis captions to “trillion pass-km or ton-km,” as recommended.

P9

L186 I typically use the word “activity” rather than “service.” Therefore, wherever you first introduce the word service, for clarity, could you add something like “service (i.e., the volume of passenger-km or ton-km) or else “service (also called “activity,” i.e., the volume of passenger-km or ton-km)?

We have incorporated this suggestion on line 164, adding a parenthetical following our mention of service: “service (i.e., transportation activity, in passenger-kilometers or ton-kilometers)...”

P10

L218 Interesting point about reduced transportation ambition forcing other sectors to limit emissions

We thank the reviewer for this comment.

P13

L277-278 Line reads that passenger increases while freight decreases, but on looking at Fig 5c passenger appears to peak around 2070 and then fall while freight rises continually, can you elaborate?

We thank the reviewer for their clarifying question. In this statement, we are commenting on the *change* in passenger and freight service in the SSP2 scenario (and, in the revised manuscript, in the SSP3 and SSP4 scenarios as well) relative to the standard 1.5°C high scenario, not on the absolute trends in passenger and freight service with time. In absolute terms, indeed passenger service peaks and then declines after about 2070 in all the scenarios (except SSP3, in which it continually rises) and freight service continually rises in all the scenarios after mid-century. However, passenger service is higher than in the standard 1.5°C high scenario in all of the SSP sensitivity scenarios, while freight service is lower than in the standard 1.5°C high scenario in SSP2-4. The overall behaviors of total passenger and freight service are largely driven by the underlying population and GDP trends, respectively (Figure S14); population peaks and declines in all SSPs except SSP3, driving the trends in passenger service, while GDP continually increases in all SSPs, driving the trends in freight service.

We have revised this sentence in the manuscript to clarify our intended message: “In the SSP2, SSP3, and SSP4 scenarios, passenger transport service demand is higher than in the standard high scenario primarily due to the larger global population in those scenarios, while

freight service demand is lower due to reduced global GDP (Figures 6c and S14)” (lines 343-345).

P16

L319 “exploration of inter- and intra-sectoral dynamics is important etc.” – this is a key point and is well illustrated in the paper

We thank the reviewer for this comment and also note that we have even further expanded our consideration of these systems dynamics throughout the paper, based on the feedback from other reviewers (see, e.g., Figures 4 and 5 and accompanying discussion).

L332 “Though electricity is the most important etc...” – because the increase in low-carbon electricity (e.g., from renewables) required is so large, this might be a good place to work in some discussion of what is needed to scale up ambition beyond a larger global financial investment. See my main comment above.

We thank the reviewer for their suggestion. As mentioned above, we have restructured our Discussion section and now highlight these issues on lines 419-449.

L335 You might mention that currently a large amount of industrial hydrogen is produced but almost entirely using steam reforming of methane (“grey hydrogen”), which emits CO2. It is in my view legitimate to use some amount of grey hydrogen in the short run while the technologies are being worked out, even as infrastructure is being developed to deliver green or blue hydrogen.

We thank the reviewer for this suggestion and have included a comment to this effect (while also noting that we revised our Discussion section, so the issue of indirect emissions from hydrogen production is discussed in a different location now) on lines 435-438: “To ensure that electric and hydrogen-based vehicles are truly zero emissions beyond the tailpipe, the upstream production of these fuels must also decarbonize^{10,63,64}; the transition from current fossil fuel-dominated production would need to occur alongside an expansion in generating capacity to meet rising electricity and hydrogen demand¹³.”

L337 “hydrogen transmission and distribution networks:” It is in my view not a certainty that a hydrogen transmission network is a requirement. This infrastructure might instead use the electricity grid to move low-carbon electricity to the point of refueling hydrogen-powered vehicles (including ships and aircraft), so as to avoid the challenges of building out a hydrogen grid.

We thank the reviewer for noting this. GCAM does incorporate a representation of both centralized hydrogen production (and transmission via either pipelines or trucks) and on-site production at service stations. We have revised this statement accordingly: “Similarly, ubiquitous transportation hydrogen use would require the large-scale development of vehicle technologies, as well as hydrogen transmission and distribution networks and/or on-site electrolysis systems at refueling stations^{10,49}.” (lines 432-435)

Reference:

Vanek, F, L Angenent, J Banks, R Daziano, and M Turnquist. (2014) Sustainable Transportation Systems Engineering. McGraw-Hill, New York.

Reviewers' Comments:

Reviewer #1:

Remarks to the Author:

The authors have done a fine job of responding to my earlier comments, as well as those of the other reviewers. The manuscript is much improved as a result.

Reviewer #2:

Remarks to the Author:

Thank you for the authors revising the manuscript responding to the reviewers' comments. While the manuscript has been much improved, I still have a few comments as follows.

- Responses to the comment in terms of novelty in Page 6

"However, an equally important ambition of our study is to offer a synthesis of a range of possible outcomes for the transportation sector in a 1.5°C-consistent future"

Thank you for the clarification. I understand the aim of this paper, and I would not strongly assert that additional novel aspects are essential in this paper, if the Editors also agree with that. However, according to the responses, the "range of possible outcome" is still unclear to me. In general, economic and environmental implications can be "possible outcome" of mitigation scenario assessment, but I consider these implications are still lacking to support the novelty of this paper. In addition, the current abstract mainly discusses about technology implications (that are already known in the existing literatures), whilst discussion on "range of possible outcome" is not included. I suggest supplementing and highlighting what exactly "range of possible outcome" is in the main text and abstract.

- Cost implications

Thank you for providing break-even carbon prices for some transport technologies. These techno-economic information may be useful, but I expected to see more general economic indicators of energy system transformation, as I have commented on it in the context of feasibility of energy system changes. Also, the authors highlighted that transport decarbonization resulted in moderate changes in other sectors, the economic information in other sectors can be useful information. For example, (economy-wide) carbon prices and policy costs (total energy system costs, GDP losses, Area under MAC curve, etc.) of each scenario are required, while I presume they are provided in the dataset deposited in Zenodo. In addition, how much energy investment is required in each scenario? Ideally technology-wise investments should be provided to inform the economic aspects of net zero transport scenarios.

Also, as Table S4 includes only the break-even prices for navigation and aviation, I would like to see them for other transport mode, such as road transport for both passenger and freight.

- Sensitivity scenarios

Thank you for the additional sensitivity scenario assessments. While these information is useful in some extent, more technology sensitivity scenarios, especially for electrification and hydrogen uses, are required. It helps support the robustness of key findings of this study in terms of the role of electrification, hydrogen and biomass as elaborated in abstract as follows.

"We highlight the leading low carbon technologies for each transportation mode, finding that electrification contributes most to decarbonization across the sector. Biofuels and hydrogen are particularly important for aviation and shipping."

As I understand that space is limited in the main text, I expect seeing them in the supplementary information document.

Reviewer #3:

Remarks to the Author:

I have read the revisions in response to my comments on the first draft, and I am satisfied with the changes. I recommend publication if the editor and other reviewers are in agreement.

REVIEWERS' COMMENTS

Reviewer #1 (Remarks to the Author):

The authors have done a fine job of responding to my earlier comments, as well as those of the other reviewers. The manuscript is much improved as a result.

We are grateful for the valuable comments from the reviewers. The manuscript is much improved thanks to the review comments.

Reviewer #2 (Remarks to the Author):

Thank you for the authors revising the manuscript responding to the reviewers' comments. While the manuscript has been much improved, I still have a few comments as follows.

We are grateful for the valuable comments from the reviewers. The manuscript is much improved thanks to the review comments.

- Responses to the comment in terms of novelty in Page 6

“However, an equally important ambition of our study is to offer a synthesis of a range of possible outcomes for the transportation sector in a 1.5°C-consistent future”

Thank you for the clarification. I understand the aim of this paper, and I would not strongly assert that additional novel aspects are essential in this paper, if the Editors also agree with that. However, according to the responses, the “range of possible outcome” is still unclear to me. In general, economic and environmental implications can be “possible outcome” of mitigation scenario assessment, but I consider these implications are still lacking to support the novelty of this paper. In addition, the current abstract mainly discusses about technology implications (that are already known in the existing literatures), whilst discussion on “range of possible outcome” is not included. I suggest supplementing and highlighting what exactly “range of possible outcome” is in the main text and abstract.

Thank you for this comment. When stating that our work synthesizes a “range of possible outcomes for the transportation sector in a 1.5°C-consistent future,” we are referring both to outcomes specific to the transportation sector and its technological transition, as well as the broader economic and environmental outcomes that the reviewer highlights. Using our high-medium-low scenario framework, we can evaluate a range of different levels of adoption of emerging low-carbon technologies within the transportation sector, and how these varied levels of technological transition can be consistent with economy-wide deep decarbonization and affect emissions from the sector. With this framework, we can also assess interconnected impacts on other sectors and outcomes for resource demands and fuel availability. We highlight this in the Introduction, specifically on page 3 writing “These varied degrees of ambition allow for an evaluation of the integrated system responses resulting from different levels of realization of climate goals in the transportation sector, spanning the range of the most aggressive decarbonization targets

that have been set by industry leaders³⁷⁻⁴⁰ (high scenario) to the minimum goals agreed upon by international organizations^{41,42} (medium and low scenarios).” Our results section on “Impacts beyond the transportation sector” discusses these outcomes in detail. We have also revised our abstract both to shorten it, as per editorial suggestions, and to further highlight these outcomes, with the final sentence in the abstract now reading “Our most ambitious scenario eliminates transportation emissions by mid-century, contributing substantially to achieving climate targets but requiring rapid technological shifts with integrated impacts on fuel demands and availability and upstream energy transitions.”

- Cost implications

Thank you for providing break-even carbon prices for some transport technologies. These techno-economic information may be useful, but I expected to see more general economic indicators of energy system transformation, as I have commented on it in the context of feasibility of energy system changes. Also, the authors highlighted that transport decarbonization resulted in moderate changes in other sectors, the economic information in other sectors can be useful information.

For example, (economy-wide) carbon prices and policy costs (total energy system costs, GDP losses, Area under MAC curve, etc.) of each scenario are required, while I presume they are provided in the dataset deposited in Zenodo. In addition, how much energy investment is required in each scenario? Ideally technology-wise investments should be provided to inform the economic aspects of net zero transport scenarios.

Also, as Table S4 includes only the break-even prices for navigation and aviation, I would like to see them for other transport mode, such as road transport for both passenger and freight.

Thank you for this comment. GCAM primarily uses the marginal abatement cost data for estimating cost of policy. We have included the carbon price data across the scenarios in our analysis in Supplementary Figures 10 and 15. We have also added breakeven prices for other transport modes, including passenger and freight road transport, to Supplementary Table 4.

- Sensitivity scenarios

Thank you for the additional sensitivity scenario assessments. While these information is useful in some extent, more technology sensitivity scenarios, especially for electrification and hydrogen uses, are required. It helps support the robustness of key findings of this study in terms of the role of electrification, hydrogen and biomass as elaborated in abstract as follows.

“We highlight the leading low carbon technologies for each transportation mode, finding that electrification contributes most to decarbonization across the sector. Biofuels and hydrogen are particularly important for aviation and shipping.”

As I understand that space is limited in the main text, I expect seeing them in the supplementary information document.

Thank you for this comment. More sensitivity analyses are always useful to add more robustness to research. However, given the space constraints, we believe the additional sensitivity analyses would need to be addressed in future studies.

Reviewer #3 (Remarks to the Author):

I have read the revisions in response to my comments on the first draft, and I am satisfied with the changes. I recommend publication if the editor and other reviewers are in agreement.

We are grateful for the valuable comments from the reviewers. The manuscript is much improved thanks to the review comments.